# Examining multiple cellular pathways at once using multiplex hextuple luciferase assaying

Alejandro Sarrion-Perdigones [1], Lyra Chang [2,3], Yezabel Gonzalez [1], Tatiana Gallego-Flores [4,7], Damian W. Young [1,2,3,5] & Koen J.T. Venken [1,2,3,5,6]*

Sensitive simultaneous assessment of multiple signaling pathways within the same cells requires orthogonal reporters that can assay over large dynamic ranges. Luciferases are such genetically encoded candidates due to their sensitivity, versatility, and cost-effectiveness. We expand luciferase multiplexing in post-lysis endpoint luciferase assays from two to six. Light emissions are distinguished by a combination of distinct substrates and emission spectra deconvolution. All six luciferase reporter units are stitched together into one plasmid facilitating delivery of all reporter units through a process we termed solotransfection, minimizing experimental errors. We engineer a multiplex hextuple luciferase assay to probe pathway fluxes through five transcriptional response elements against a control constitutive promoter. We can monitor effects of siRNA, ligand, and chemical compound treatments on their target pathways along with the four other probed cellular pathways. We demonstrate the effectiveness and adaptiveness of multiplex luciferase assaying, and its broad application across different research fields.

[1] Verna and Marrs McLean Department of Biochemistry and Molecular Biology, Baylor College of Medicine, Houston, TX 77030, USA. [2] Department of Pharmacology and Chemical Biology, Baylor College of Medicine, Houston, TX 77030, USA. [3] Center for Drug Discovery, Baylor College of Medicine, Houston, TX 77030, USA. [4] Department of Molecular and Human Genetics, Baylor College of Medicine, Houston, TX 77030, USA. [5] Dan L. Duncan Comprehensive Cancer Center, Baylor College of Medicine, Houston, TX 77030, USA. [6] McNair Medical Institute at The Robert and Janice McNair Foundation, Baylor College of Medicine, Houston, TX 77030, USA. [7] Present address: Max Planck Institute for Brain Research, Frankfurt am Main 60438, Germany. *email: koen.j.t.venken@gmail.com

Currently, most cell-based screening assays rely on a single measurement (*e.g.*, enzyme inhibition or cell viability) to identify genetic or pharmacological agents that modulate the biomedical phenomenon of interest[1]. However, due to the complexity of biological systems, screens based on single biological measurements yield only limited information[2,3]. Additional screening assays must be performed to expand information content, but this often requires protracted assay development, time, and/or expense. Moreover, independently performed screens may not always be appropriately controlled, making comparative analysis across different experimental variables difficult[4,5].

Multiplexed cellular screens seek to address these limitations by measuring multiple readouts from a single screening unit simultaneously. Multiplexed biological screening from a single sample can provide a more detailed cellular signature, allowing greater perspective into the nuances which differentiate normal versus disease-associated processes[6–8]. Moreover, multiple simultaneous measurements of the same sample enable correlation and comparison between experimental variables and biological effects[9]. Some examples of multiplexed biological screening already exist that rely on fluorescent labeling. Modern flow cytometry can analyze the presence of up to 30 different antigens using labeled antibodies[10], whereas cell painting measures the presence of a multitude of morphological features of cells using a combination of fluorescent dyes and automated image analysis[11]. On the other hand, to accommodate the sensitive reporting of subtle changes in signaling pathway activities over large dynamic ranges, luciferase reporters have emerged as a promising addition to the multiplex biological screening toolbox[12].

Luciferase reporters are widely used in biomedical research for a variety of applications, including gene expression, intracellular signaling, transcription factor characterization, receptor activity, protein folding, drug screening, analytics, and immune-based assays[12,13]. Luciferases have many advantages over fluorescent proteins, such as higher sensitivity, wider dynamic detection range[13,14], and the absence of auto-luminescence in mammalian cells[13]. Luciferases are categorized by their enzymatic activities[13,15]: the beetle luciferases first activate their substrate (D-Luciferin and derivatives) to a product whose oxidation results in light emission, whereas the marine luciferases directly oxidize their substrate (coelenterazine, vargulin, or furimazine), generating an excited-state molecule that emits a photon. Similar to fluorescent proteins, each luciferase has a unique emission spectrum[13,16] that enables detection of two or more luciferases simultaneously as long as the spectra are distinguishable[14,15,17,18].

Most multiplex luciferase experiments today measure the light emitted from two luciferases in succession[19]. The first, which is typically FLuc, a firefly (*Photinus pyralis)* luciferase, utilizes D-Luciferin and is used to monitor a cellular signaling event of interest. The second enzyme, often the Renilla luciferase of *Renilla reniformis* (Renilla) that uses coelenterazine[20] or the brighter NLuc (a synthetic version of the *Oplophorus gracilirostris* luciferase) that uses furimazine[21], is used as an internal control. These dual-luciferase assays allow sequential quantitative measurements of both luciferase activities in the same sample, thereby eliminating pipetting errors that could occur if measurements were performed from different samples[22]. In recent years, researchers have also developed their own (non-commercial) versions of these assays with presumably similar performance[23,24].

Inspired by the urgent need to report on subtle changes in multiple signaling pathway activities over large dynamic ranges, and the variety in substrate and spectral properties inherent to natural luciferase enzymes, we decided to explore multiplex capabilities for luciferase-based reporter systems. Here we report

multiplexed luciferase reporter assaying, capable of measuring six parameters within the same experiment, called multiplex hextuple luciferase assaying. We demonstrate the utility of this assaying by simultaneously monitoring the direct and collateral effects of introducing siRNAs, pharmaceutical drugs, and ligands targeted toward specific pathways. Our multiplex hextuple luciferase assaying is adaptable to most cellular signaling pathways of interest, cost-effective, easy to integrate into any research setting already using the dual-luciferase assay, and offers the immediate implementation in large-scale multiplex drug screening efforts.

## Results

**Concept of multiplex luciferase assaying**. The commonly used dual-luciferase assay orthogonally detects FLuc and Renilla through sequential application of their unique substrates, D-Luciferin and coelenterazine, with proper quenching of light emission from the first enzyme before the second substrate is introduced. This is accomplished through the addition of buffers at specific time points during the assay[22]. Cells are usually co-transfected with two plasmids, one encoding a control luciferase reporter and the other encoding a pathway-specific luciferase reporter, are washed and lysed before addition of D-Luciferin and subsequent measurement of its emission. Then a quencher is added in the presence of coelenterazine, and its emission is measured (Supplementary Fig. 1a). Based on this paradigm, we explored different options for expanding the number of luciferases catalyzing each sequential stage using readily available reagents (Supplementary Fig. 1b).

**Criteria for luciferases in multiplex luciferase assaying**. We used four criteria to identify luciferases that might be orthogonally detected in a single multiplex luciferase experiment: (1) preference for consumption of a single substrate over any other, and potential for luminescence quenching, (2) in cases where the same substrate is preferred, they must exhibit minimally overlapping emission spectra that can be distinguished by emission filters and/or mathematical computation, (3) stable light emission throughout the experiment to ensure high quality measurements, and (4) a wide dynamic range for light emission to be able to detect subtle differences in pathway activities. We assessed these four criteria for 12 luciferases: 6 have been reported to emit light using D-Luciferin as a substrate, 1 to prefer furimazine, and 5 to prefer coelenterazine (Table 1).

**Single substrate consumption and luminescence quenching**. To determine the substrate preference of the 12 luciferases (Table 1), transcriptional units containing each luciferase gene were constructed using multipartite assembly synthetic biology[25–27] (Fig. 1a). To ensure adequate expression levels following transfection, the human cytomegalovirus (hCMV-IE1) promoter was placed upstream of each luciferase gene, and the bovine growth hormone poly-adenylation terminator (bGHpA) was cloned downstream[28]. Vectors containing each assembled luciferase transcriptional unit were transfected into HEK293T/17 cells, and 24 h after transfection, the absolute luciferase activity was measured after cell lysis in an endpoint experiment. Substrate preference and emission brightness of each luciferase were characterized after addition of D-Luciferin-containing buffer alone, or followed by the addition of a second buffer, containing quencher and coelenterazine. Luciferases that prefer coelenterazine as a substrate are generally brighter than those that prefer D-Luciferin so we diluted 10 times the lysates of cells expressing the coelenterazine-specific luciferases to prevent detection saturation. In total 6 of the 12 luciferases were responsive to D-Luciferin as expected; however, some, albeit significantly reduced,

**Table 1 List of luciferases evaluated in this study.**

| Abbreviation | Full name | Luciferase and origin | Substrate | Emission peak (nm) | Reference |
|---|---|---|---|---|---|
| ELuc | Enhanced Beetle Luciferase | Green luciferase from *Pyrearinus termitilluminans* | D-Luciferin | 537 | 64 |
| CBG | Click Beetle Green Luciferase | Green luciferase from *Pyrophorus plagiophthalamus* | D-Luciferin | 537 | 65 |
| RoLuc | *R. ohbai* Luciferase | Green luciferase from *Rhagophthalmus ohbai* | D-Luciferin | 557 | 18 |
| FLuc | Firefly Luciferase | Firefly luciferase from *Photinus pyralis* (luc2 version) | D-Luciferin | 562 | 66 |
| RedF | Red Firefly Luciferase | Red mutant S286Y luciferase from *Luciola curciata* | D-Luciferin | 614 | 67 |
| RedLuc | *P. hirtus* Red Luciferase | Red luciferase from *Phrixothrix hirtus* | D-Luciferin | 617 | 68 |
| NLuc | Nano Luciferase | Directed evolved synthetic NanoLuc luciferase, from *Oplophorus gracilirostris* | Furimazine | 462 | 69 |
| Renilla | Renilla Luciferase | Luciferase from *Renilla reniformis* | Coelenterazine | 481 | 70 |
| MetLuc | *Metridia* Luciferase | Luciferase from *Metridia longa* | Coelenterazine | 482 | 71 |
| Lucia | Lucia Luciferase | Synthetic luciferase from Invivogen | Coelenterazine | 486 | 72 |
| GLuc | *Gaussia* Luciferase | Luciferase from *Gaussia princeps* | Coelenterazine | 487 | 73 |
| GrRenilla | Green Renilla Luciferase | Synthetic mutant Green Renilla luciferase from *Renilla reniformis* | Coelenterazine | 532 | 74 |

luminescence levels remained for some of the D-Luciferin-specific luciferases after addition of quencher and coelenterazine. The other six enzymes were responsive to coelenterazine, and showed minimal promiscuous activity with D-Luciferin and background levels were empirically set at $10^3$ RLU/s (Fig. 1b). Although NLuc has been engineered to use the substrate furimazine, it showed strong activity with coelenterazine.

Only D-Luciferin-responsive luciferases that can be quenched down to levels similar to FLuc during the second measurement should be incorporated into multiplex luciferase assaying. The three D-Luciferin luciferases that showed the best signal quenching in terms of relative emission were ELuc (98%), FLuc (99%), and RedF (99.3%) (Fig. 1c). Although the D-Luciferin signal of ELuc was only quenched to 98% of its original value, ELuc exhibited a lower absolute luminescence than FLuc and RedF (Fig. 1b). Nevertheless, all three enzymes emitted negligible light after the addition of coelenterazine substrate compared to the background level of $10^3$ RLU/s (Fig. 1b). Our substrate specificity testing identified nine luciferases to be evaluated against the second criterion, namely three D-Luciferin-responsive luciferases (ELuc, FLuc, and RedF) and six coelenterazine-responsive luciferases (NLuc, Renilla, MetLuc, Lucia, GLuc, and GrRenilla).

**Ability to deconvolute emission spectra.** To determine regions of minimal emission overlap between luciferases belonging to a single substrate group, emission spectra of each luciferase were recorded (Supplementary Fig. 2a). ELuc, FLuc, and RedF have distinct emission spectra with maxima at 537 nm, 562 nm, and 617 nm, respectively that can be exploited in multiplexing experiments (Fig. 1d and Supplementary Fig. 2b). However, It would be impractical to attempt to distinguish between the emission spectra of the coelenterazine-responsive luciferases (NLuc, Renilla, MetLuc, Lucia, GLuc, and GrRenilla) since Renilla, MetLuc, Lucia, and GLuc exhibit maximum emission within 7 nm of each other (482–487 nm) (Fig. 1e and Supplementary Fig. 2c). However, NLuc (462 nm) and GrRenilla (532 nm) could be readily distinguished from any of the other four, including Renilla (481 nm). Therefore, we decided to proceed with only three coelenterazine-responsive luciferases, namely NLuc, Renilla, and GrRenilla (Fig. 1f and Supplementary Fig. 2c).

We chose two bandpass (BP) filters to deconvolute a multiplexed luminescence measurement of the three D-Luciferin-responsive luciferases: BP515-30, which measures between 500 and 530 nm, and BP530-40, which measures between 510 and 550 nm (Fig. 1d). We calculated two transmission coefficients ($\kappa$) for each luciferase by dividing the light that transmits through either filter (*e.g.*, $ELuc_{515}$ for BP515-30) by the total light emitted by each of the luciferases (*e.g.*, $\kappa ELuc_{515} = ELuc_{515}/ELuc_{TOTAL}$) (Supplementary Fig. 3a, b). Using the six transmission coefficients ($\kappa ELuc_{515}$, $\kappa FLuc_{515}$, $\kappa RedF_{515}$, $\kappa ELuc_{530}$, $\kappa FLuc_{530}$, and $\kappa RedF_{530}$) (Supplementary Fig. 3c), we established a mathematical model that consists of two simultaneous equations, each with three equations and three variables, to determine the contribution of the three individual luciferases to the experimental luminescence value recorded with the BP515-30 ($Light_{515}$) and BP530-40 filters ($Light_{530}$), as well as without any filter ($Light_{TOTAL}$) (Supplementary Fig. 4a). Using this model, total luminescence was calculated for ELuc, FLuc, and RedF (Supplementary Fig. 4b).

We used two alternative BP filters to specifically separate the spectra of coelenterazine-responsive luciferases due to the similar transmission coefficients when BP515-30 and BP530-40 are used for NLuc and Renilla: BP410-80, which measures between 370 and 450 nm, and BP570-100, which measures between 520 and 620 nm (Fig. 1f). Again, we calculated the transmission coefficients ($\kappa$) by dividing the light transmitted from each luciferase through both filters ($NLuc_{410}$, $Renilla_{410}$, and $GrRenilla_{410}$, as well as $NLuc_{570}$, $Renilla_{570}$, and $GrRenilla_{570}$) by the total light emitted from each luciferase ($NLuc_{TOTAL}$, $Renilla_{TOTAL}$, and $GrRenilla_{TOTAL}$) (Supplementary Fig. 3a, d). Using the six transmission coefficients for this series ($\kappa NLuc_{410}$, $\kappa Renilla_{410}$, $\kappa GrRenilla_{410}$, $\kappa NLuc_{570}$, $\kappa Renilla_{570}$, and $\kappa GrRenilla_{570}$) (Supplementary Fig. 3e), we established a second mathematical model to determine the contribution of the three individual coelenterazine-responsive luciferases to the experimental luminescence value recorded with the BP410-80 ($Light_{410}$) and BP570-100 filters ($Light_{570}$), as well as without any filter ($Light_{TOTAL}$) (Supplementary Fig. 4c). Using this second model, we calculated the total luminescence for NLuc, Renilla, and GrRenilla (Supplementary Fig. 4d).

We expressed the six luciferases in different human cell lines (A549, MCF7, MDA-MB-231, and SK-BR-3) to determine the consistency of the transmission coefficients (Supplementary Table 1). Data from these experiments demonstrate that the transmission coefficients are reproducible between cell lines (Supplementary Table 1). Nonetheless, re-evaluation of the transmission coefficients in every experimental setup is recommended before performing experimental assays since changes in pH or temperature can slightly affect the luciferase emission spectra and therefore the transmission coefficient values[29].

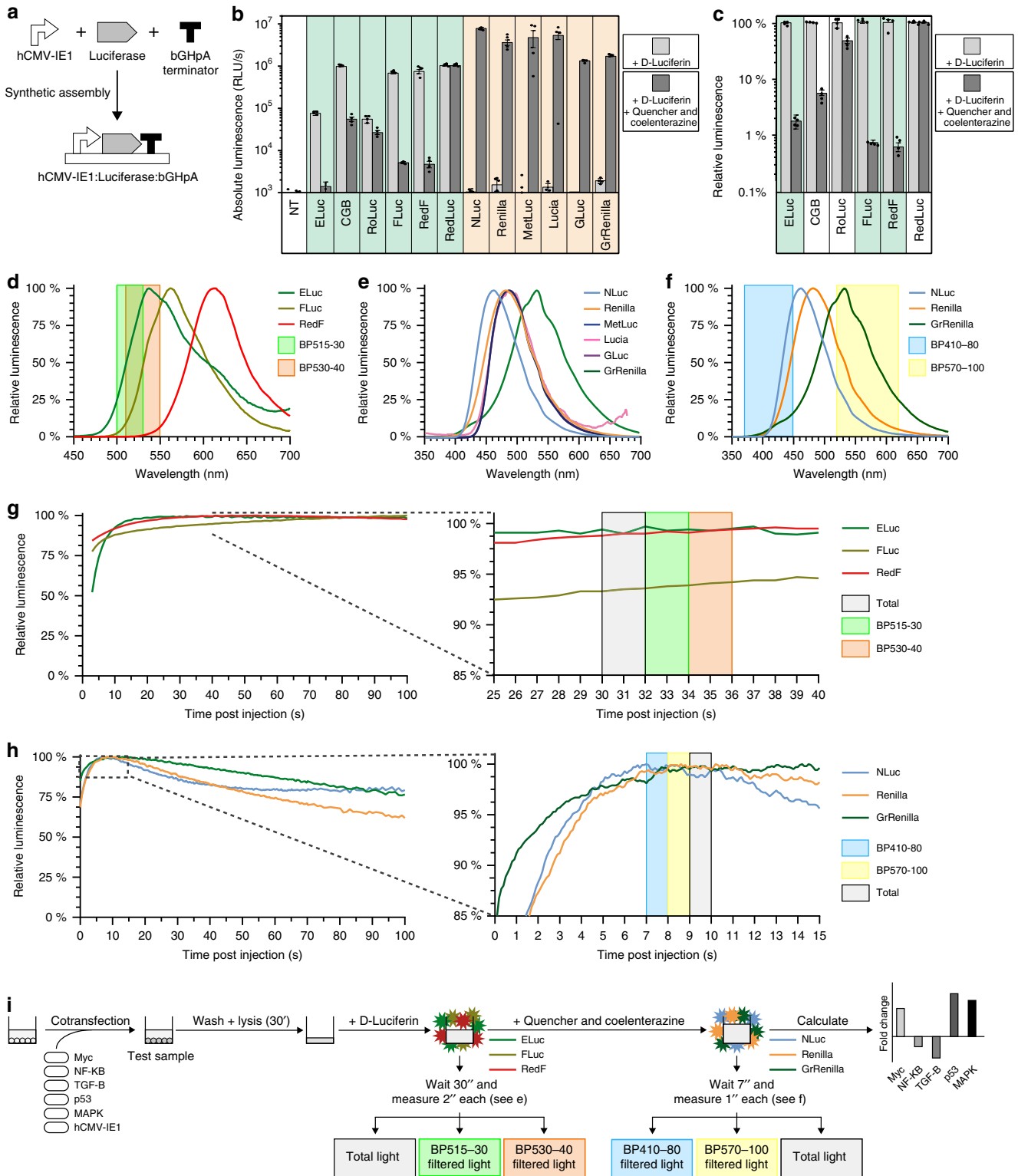

**Stable light emission throughout the experiment**. To determine the best time window to perform the experiment, light emission was recorded for 180 s after substrate injection. ELuc, FLuc, and RedF are glow-type luciferases that exhibit a prolonged stable signal but with lower intensity (Fig. 1g, left, and Supplementary Fig. 5a, b–e, top, left). ELuc plateaued after 12 s and remained very stable over time, but FLuc and RedF required more time for the luminescence signal to stabilize. In fact, FLuc emission did not plateau in the 180 s after substrate injection despite a slight

change in signal after 25 s. RedF was very stable between 20 and 70 s and then decayed slowly. The optimal period for measuring these luciferases was determined to be at 30 s post-injection (Fig. 1g, right, and Supplementary Fig. 5e, right). Since the intensity of D-Luciferin-responsive luciferases is lower compared to the coelenterazine-responsive ones, we proceeded with 2-second measurement windows.

In contrast, NLuc, Renilla, and GrRenilla are flash-type luciferases that elicit a stronger yet shorter-lived signal (Fig. 1h,

**Fig. 1 Experimental parameters for multiplex hextuple luciferase assaying. a** Schematic of obtaining constitutively expressed luciferase transcriptional units by synthetic assembly of a constitutive hCMV-IE1 promoter, luciferase, and bovine growth hormone polyadenylation (bGHpA) terminator. **b** Absolute luminescence emitted by 12 transcriptional luciferase units during both steps of a dual-luciferase assay. Luminescence was recorded after adding solely D-Luciferin, or additional quencher and coelenterazine. RLU/s relative light units per second, NT non-transfected control. c Quenching potency of D-Luciferin luciferases after adding solely D-Luciferin, or additional quencher and coelenterazine. To visualize the quenching potency for the D-Luciferin luciferases absolute luminescence (see **b**) is represented in relative units. **d** Emission spectra of three quenchable and spectrally distinguishable D-Luciferin luciferases. Two bandpass emission filters, measuring between 500 and 530 nm (BP515-30) and 510 and 550 nm (BP530-40), were used for spectral unmixing, and are indicated over the spectra. **e** Emission spectra of six coelenterazine luciferases. Spectral overlap between Renilla, MetLuc, Lucia, and GLuc illustrates likely interference with spectral unmixing. **f** Emission spectra of the three most optimal spectrally distinguishable coelenterazine luciferases. Two bandpass emission filters, measuring between 370 and 450 nm (BP410-80) and 520 and 620 nm (BP570-100), were used for spectral unmixing, and are indicated over the spectra. **g** Time interval to perform emission measurements after adding D-Luciferin. Overlay of the kinetic charts of the three D-Luciferin luciferases (left) and a close-up of the section between 25 and 40 s (right) reveal three 2-s intervals, 30 s post-injection, to perform measurements. **h** Time interval to perform emission measurements after adding D-Luciferin, quenching agent and coelenterazine. Overlay of the kinetics of the three coelenterazine luciferases (left) and a close-up window of the section between 0 and 15 s (right) reveal three 1-s intervals, 7 s post injection, to perform measurements. **i** Schematic of the empirically determined multiplex hextuple luciferase assay. After cotransfection, cells are incubated for 24 h. Next, cell samples are washed, lysed for 30 min, and transferred to a plate reader equipped with appropriate filters (**d**, **f**). Subsequently, D-Luciferin is added and three emission measurements are recorded 30 s later: total light, BP515-30-filtered light, and BP530-40-filtered light (**g**). Finally, quencher and coelenterazine are added and three additional emission measurements are recorded 7 s later: BP410-80-filtered light, BP570-100-filtered light, and total light (**h**) (Supplementary Fig. 7). Four technical replicates are included in each data point, and the standard error of the mean is represented (**b**, **c**). Spectra **d**–**f** include data from one spectral scan, showing no significant changes when compared with four other technical replicates. Kinetic charts **g**–**h** include data from one biological sample, showing no significant changes when compared with four other technical replicates. Source data are provided as a Source Data file.

left, and Supplementary Fig. 6a-e, left). We determined that the best window for measuring these luciferases was at 7 s post-injection because the luminescence of NLuc decays quickly after 10 s (Fig. 1h, right, and Supplementary Fig. 6e, right). Due to their strong luminescence, only 1-second measurements were needed for these luciferases.

It is important to note that quenching kinetics of the D-Luciferin-responsive luciferases must be fast to minimize background during the second step of the assay. Fortunately, the luminescence dropped to background levels within 4 s post-injection and did not affect measurements of the coelenterazine-responsive luciferases (Supplementary Fig. 5b-d, bottom). Based on all of these results, we established a protocol for multiplex hextuple luciferase assaying (Fig. 1i and Supplementary Fig. 7). Briefly, cells are washed and lysed at 24–48 h post-transfection of luciferase reporter plasmids. Cell lysates are then incubated with D-Luciferin substrate for 30 s before total light, BP515-30-filtered light, and BP530-40-filtered light are measured serially (2 s/ measurement). Quencher and coelenterazine substrate are then added, and cell lysates are incubated for 7 s before BP510-80-filtered, BP570-100-filtered, and total light are measured serially (1 s/measurement). At the end of this protocol, six luminescence values are available for further analysis.

**Wide dynamic range of light emission.** Next, we sought to determine whether simultaneous quantification of three luciferases is possible across a wide dynamic range. To accomplish this, we determined the quantitative relationships between the three luciferases within each group using serial dilutions of the cell lysates. We transfected ELuc, FLuc, or RedF-expressing vectors into HEK293/T17 cells. Seven cell lysate mixtures with defined volume ratios (0:100, 20:80, 40:60, 50:50, 60:40, 80:20, and 100:0 ratios) of two of the three luciferases were prepared in addition to lysates consisting of an equal volume of the third luciferase. After addition of D-Luciferin substrate to each mixture, total light, BP515-30-filtered light, and BP530-40-filtered light were measured (Fig. 2a, and Supplementary Fig. 8a). The deconvoluted luminescence signals of ELuc and FLuc were in proportion with their relative amounts added to the mixture and their respective regression curves showed excellent linearity ($r^2_{ELuc} = 0.9982$ and $r^2_{FLuc} = 0.9998$) (Fig. 2b and Supplementary

Fig. 8b, left). The RedF luciferase signal was constant in all the samples, and the slope of this regression did not significantly differ from zero as expected. We performed a similar experimental setup for the two other combinations, resulting in similar quality regression lines when FLuc and RedF were varied and ELuc was kept constant ($r^2_{FLuc} = 0.9906$ and $r^2_{RedF} = 0.9993$) (Fig. 2c and Supplementary Fig. 8c, left), as well as when ELuc and RedF were varied and FLuc was kept constant ($r^2_{ELuc} = 0.9979$ and $r^2_{RedF} = 0.9979$) (Fig. 2d and Supplementary Fig. 8d, left). Next, we determined the dynamic range of luciferase separation using serial dilutions of the seven mixtures and found it to be stable over at least two orders of magnitude spanning from $>10^6$ to $\geq 10^4$ RLU/s for the D-Luciferin-responsive luciferases, demonstrating that excellent linearity was maintained (Supplementary Fig. 8b-d). Importantly, the transmission coefficients did not change in the proposed experimental dynamic range (Supplementary Table 2).

We performed a similar experimental setup for the coelenterazine-responsive luciferases, adding the appropriate quencher and coelelenterazine solution (Fig. 2e and Supplementary Fig. 9a). These experiments showed good linearity in all possible combinations: when NLuc and Renilla were varied and GrRenilla was kept constant ($r^2_{NLuc} = 0.9868$ and $r^2_{Renilla} = 0.9979$) (Fig. 2f and Supplementary Fig. 9b, left), when Renilla and GrRenilla were varied and NLuc was kept constant (i.e., $r^2_{Renilla} = 0.9825$ and $r^2_{GrRenilla} = 0.9996$) (Fig. 2g and Supplementary Fig. 9c, left), as well as when NLuc and GrRenilla were varied and Renilla was kept constant ($r^2_{NLuc} = 0.9976$ and $r^2_{GrRenilla} = 0.9877$) (Fig. 2h and Supplementary Fig. 9d, left). Similar to the D-Luciferin-luciferases, the dynamic range of separation of these luciferases was stable over at least two orders of magnitude spanning from $>10^7$ to $\geq 10^5$ RLU/s, demonstrating that linearity was maintained (Supplementary Fig. 9b-d). Again, the transmission coefficients for the coelenterazine- luciferases did not change in the proposed experimental dynamic range either (Supplementary Table 2).

Collectively, these results indicate that the activities of the three enzymes within each luciferase group can be determined accurately and deconvoluted with a dynamic range of $>10^6$ to $\geq 10^4$ RLU/s for the D-Luciferin- luciferases and $>10^7$ to $\geq 10^5$ RLU/s for the coelenterazine luciferases, using the established protocol (Fig. 1i and Supplementary Fig. 7).

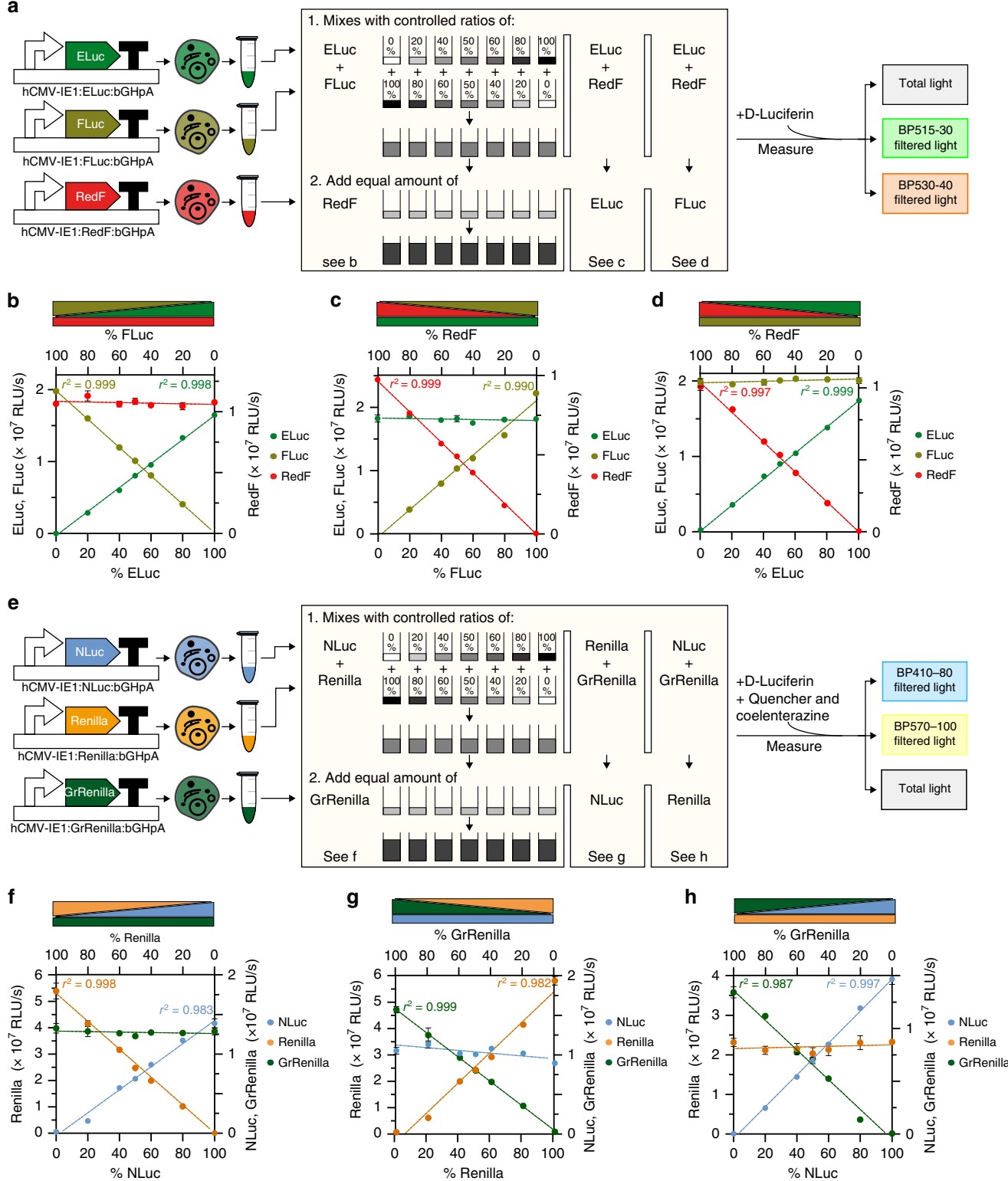

**Pipeline to clone an all-in one multiplex luciferase vector**. We implemented a synthetic assembly cloning approach to generate multigenic vectors in a rapid and adaptable fashion to favor stoichiometric cellular uptake of each reporter in each transfected cell. As a proof of concept, we designed a multiplex hextuple luciferase vector to simultaneously probe transcriptional signaling through c-Myc, NF-κβ, TGF-β, p53, and MAPK/JNK response elements (RE) against a control constitutive promoter, the

hCMV-IE1 promoter[28]. We tested several constitutive promoters (hCMV-IE1, hEF1A, PGK, and SV40), under varying experimental conditions and empirically found that the hCMV-IE1 promoter was the most consistent in expression levels amongst different cell lines. Synthetic assemblies were performed using the GoldenBraid 2.0 platform[25,26], which stitches DNA fragments together using Type IIs restriction enzymes and unique 4-base pair overhangs to direct the order of assembly of the different

**Fig. 2 Quantitative determination of the dynamic range within luciferase mixtures. a** Schematic of the experimental setup used to confirm the quantitative relationships between the D-Luciferin-responsive luciferases in a single emission recording experiment. Individual plasmids, each having one transcriptional D-Luciferin-responsive luciferase unit (see Fig. 1a), were transfected into HEK293T/17 cells. Transfected cells were harvested and lysed after 24 h. To confirm a quantitative relationship between ELuc and FLuc **b**, defined amounts of each cell lysate were mixed at different ratios totaling 100% before addition of an equal amount of RedF. After the addition of D-Luciferin substrate, total and filtered light were measured after 30 s (Fig. 1g). Similar experimental setups were used to confirm the quantitative FLuc/RedF **c** and ELuc/RedF **d** relationships. **e** Schematic of the experimental setup used to confirm quantitative relationships between the coelenterazine-responsive luciferases in a single emission recording experiment. Individual plasmids, each having one transcriptional coelenterazine-responsive luciferase unit (see Fig. 1a), were transfected into HEK293T/17 cells. Transfected cells were harvested and lysed after 24 h. To confirm a quantitative relationship between NLuc and Renilla **f**, defined amounts of each cell lysate were mixed at different ratios totaling 100% before addition of an equal amount of GrRenilla. After the addition of quencher and coelenterazine substrate, total and filtered light were measured after 7 s (Fig. 1h). Similar experimental setups were used to determine the quantitative Renilla/GrRenilla **g** and NLuc/GrRenilla **h** relationships. For **b–d** and **f–h**, *P*-value < 0.0001 for all regression lines at varying concentrations. For luciferases kept at constant concentration, all minimal slopes that were interpolated by regression did not significantly differ from zero. Four technical replicates are included in each data point, and the standard error of the mean is represented. Source data are provided as a Source Data file.

fragments (Fig. 3a and Supplementary Fig. 10a). The DNA fragments used in this work were: (i) the hCMV-IE1 promoter[28] (ii) tandem repeats of different DNA operator elements that report on p53 (2xp53_RE), TGF-β (4xTGF-β_RE), NF-κβ (5xNF-κβ_RE), c-Myc (5xE-box_RE), and MAPK/JNK signaling (6xAP-1_RE) (Supplementary Table 3), (iii) a mini-promoter (MiniP) that was previously shown to be inactive without added enhancer elements[30], (iv) luciferase-coding DNA sequences (ELuc, FLuc, RedF, NLuc, Renilla, or GrRenilla), (v) a strong transcriptional terminator (the bovine growth hormone polyadenylation signal)[28], and (vi) a transcription blocker[31] (Fig. 3b). The control luciferase reporter unit consisted of the hCMV-IE1 promoter, ELuc, and bGH polyadenylation signal (Fig. 3c, Supplementary Fig. 10b, and Supplementary Table 4).

Different luciferase combinations were pilot-tested to ensure good measurements in multiplex luciferase assaying. NLuc was the brightest among the luciferases described here and therefore our first choice to be used as the standard, since the brightest of the FLuc/Renilla pair is typically used as normalization agent. However, when any of the coelenterazine-responsive luciferases were used as the standard, high experimental errors between biological replicates were observed after emission values for the individual luciferases were deconvoluted from the mix (Supplementary Fig. 11). Our previous results suggested that signal separation is more optimal when all luciferases emit light within a similar range (Fig. 2a–h, Supplementary Fig. 8a–d, and Supplementary Fig. 9a–d). We tested ELuc, FLuc, and RedF as possible references for normalization in the multigenic vector (Supplementary Fig. 12a). ELuc exhibited good reproducibility (Supplementary Fig. 12b–d), and was the only luciferase found to be useful as a normalization standard in multiplex luciferase assaying.

Next, we built pathway reporters from specific DNA operator elements, the miniP mini-promoter, one of five remaining luciferases, and the bGH polyadenylation signal (Fig. 3d and Supplementary Fig. 10c). We insulated the pathway reporters by adding a transcription blocker[31] (Fig. 3e and Supplementary Fig. 10d). Five insulated response pathway reporter units and the control reporter unit were sequentially stitched together to generate the multiplex hextuple luciferase reporter (Fig. 3f and Supplementary Fig. 13). Using this construct, we can simultaneously monitor changes in the activity of five signaling pathways by measuring the light emission from the individual luciferases relative to the control luciferase driven by the constitutive promoter. We performed Sanger sequencing, restriction enzyme fingerprinting, and uncut supercoiled conformation analysis to ensure that every plasmid generated was correct, intact, and stable (Fig. 3g, h and Supplementary Fig. 14).

To distinguish classical cotransfection methods from that which introduces multiple genetic elements carried on a single plasmid constructed by synthetic assembly cloning, we propose the term solotransfection. Our multiplex construct incorporates all six transcription units on the same vector, ensuring an identical copy number of each unit in every transfected cell. In contrast, the classical cotransfection method relies on equal plasmid uptake by every cell, generating greater variability between (Fig. 4a). We compared both transfection methods by solotransfecting the multigenic vector in one sample and cotransfecting a similar number of molecules for each of the six individual plasmids in a second cell sample (Supplementary Table 5). Luciferase measurements were performed at 24 h post-transfection and absolute luminescence values for the six luciferases (*n* = 4) were plotted for both methods in three different cell lines (A549, HEK293T/17, and SK-BR-3) (Fig. 4b–d).

The trend displayed by the six luciferases was similar for both approaches across the three cell lines. However, luminescence in the solotransfection experiment exhibited a smaller coefficient of variation (%CV), demonstrating that solotransfection of all reporter units assembled onto one vector leads to more consistent results than cotransfection of six individual plasmids expressing a single reporter unit. This trend was not only observed while comparing cotransfection versus solotransfection of all six transcriptional units. Cotransfecting even just two, or three, four, five and again six transcriptional units, resulted in larger coefficients of variations compared to solotransfection of the six transcriptional units (Supplementary Fig. 16a). This indicates that, besides simplifying the transfection process, the overall error of the experiment will be reduced when all units are solotransfected, ensuring robust experimental rigor and reproducibility.

**SiRNAs expose collateral effects using reporter multiplexing.** To test our multiplex luciferase approach, we analyzed the effects of previously verified siRNA knockdown on key upstream pathway-associated transcriptional response elements included in the multiple luciferase vector (Supplementary Table 6). We treated A549 cells in 96-well plates with 10 nM siRNA and incubated them for 24 h prior to solotransfection with the multiplex luciferase vector. After incubation for an additional 24 h, cells were lysed and the multiplex luciferase assay was performed as described (Fig. 5a). In parallel, A549 cells were similarly treated in 6-well plates, followed by extraction of mRNA and quantitative PCR (qPCR) to determine the effect of each siRNA on transcript levels of downstream genes regulated by each of the five signaling pathways: *CDKN1A/p21* and *BAX1* (p53 pathway), *DAPK1* and *SMAD7* (TGF-β pathway), *IL6*, *BCL-X* and *CCL2* (NF-κB pathway), *E2F1* and *TERT* (c-Myc pathway), and *MMP1* and *VEGFD* (MAPK/JNK pathway). qPCR analysis confirmed that all siRNAs reduced the target mRNA level up to 16-fold (see Fig. 5b–f,

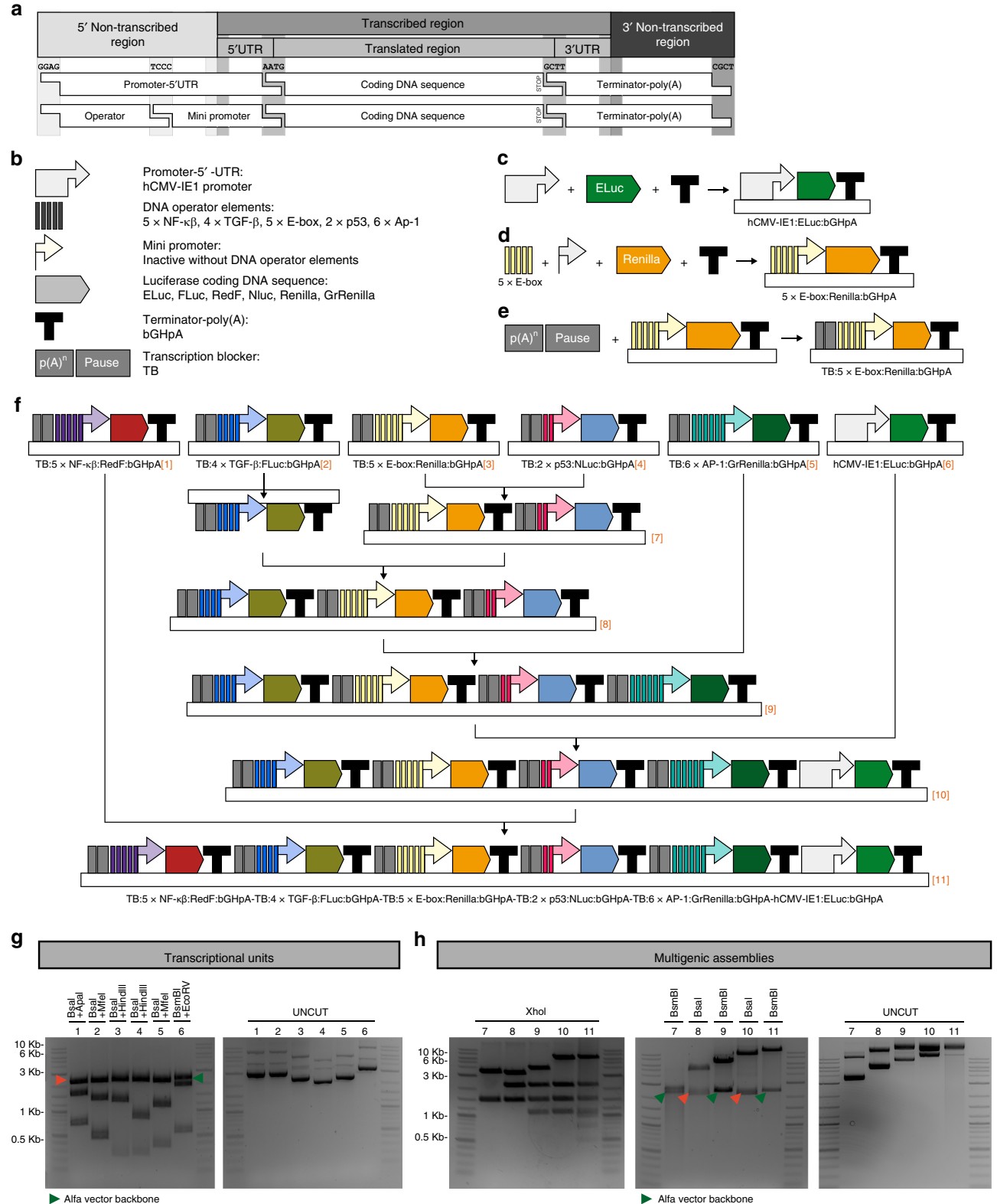

Supplementary Figs. 15, 16c-g, and Supplementary Table 7). It must be noted that the genes that were selected as downstream indicators of pathway modulation are likely regulated by multiple transcription factors due to crosstalk between the different pathways so not all genes related to a signaling pathway follow the downstream trend reported by synthetic response elements. For instance, *CDKN1A* transcription is coregulated positively by

both the p53 and TGF-β pathways[32], whereas *VEGFD* transcription is downregulated by TGF-β[33], but upregulated by MAPK/JNK[34].

The multiplex luciferase assay showed that knockdown of *TP53* transcript levels resulted in a 4-fold reduction in p53 pathway activity with significant collateral repression of the NF-κβ and MAPK/JNK pathways and activation of the c-Myc pathways. We

**Fig. 3 Implementation of synthetic assembly cloning for inserting multiple luciferase reporters into a single vector. a** Schematic of the different DNA element categories. Defined synthetic assembly cloning overhangs generated by BsaI cutting (GGAG, TCCC, AATG, GCTT, and CGCT) allow directional assembly of pre-made DNA fragments into defined transcriptional units. **b** Categories of DNA elements include: (1) Full promoter and 5′UTR (cytomegalovirus promoter, hCMV-IE1 promoter) used in all constitutively expressed transcriptional units; (2) DNA operator elements (different DNA pathway response elements whose activities are regulated by cellular signaling events, see Supplementary Table 3); (3) minimal promoter (synthetic minimal TATA-box promoter with low basal activity, mini promoter) needed for transcription initiation driven by the different DNA pathway operator response elements; (4) luciferase-coding DNA sequence (see Table 1), including codon for translation termination (STOP); (5) terminator-poly(A) sequence (terminator of the bovine growth hormone polyadenylation signal, bGHpA); and (6) transcription blocker (TB), consisting of a synthetic polyA terminator (p(A)n) and the RNA polymerase II transcriptional pause signal from the human α2 globin gene (Pause), to prevent transcriptional interference between different pathway-responsive luciferase transcriptional units. **c** Multipartite assembly of a constitutively expressed luciferase. **d** Multipartite assembly of DNA operator elements, minimal promoter, luciferase, and terminator into a pathway-responsive luciferase transcriptional unit. **e** Binary assembly of a transcription blocker (TB) upstream of each pathway-responsive unit. **f** Overview of the binary assembly steps used to stitch together six luciferase transcriptional units into a single multi-luciferase plasmid using successive bipartite assembly steps. The final multi-luciferase plasmid includes the six luciferase transcriptional units, *i.e.*, five insulated pathway-responsive luciferase transcriptional units and one constitutively expressed luciferase transcriptional unit used as the control for normalization. **g**, **h** DNA analysis (restriction enzyme fingerprinting and uncut) of all six individual luciferase transcriptional units (Plasmids 1-6) **g**, and intermediate assemblies (Plasmids 7–10) and final hextuple luciferase vector (Plasmid 11) **h**, demonstrates the stability and integrity of all plasmids. Plasmid maps and restriction enzymes used are indicated in Supplementary Fig. 14. Plasmids encoding DNA building blocks, individual luciferase transcriptional units, and the final multigenic luciferase reporter have been deposited at Addgene (https://www.addgene.org/) (Supplementary Table 4). Source data representing all agarose gel pictures are provided as a Source Data file.

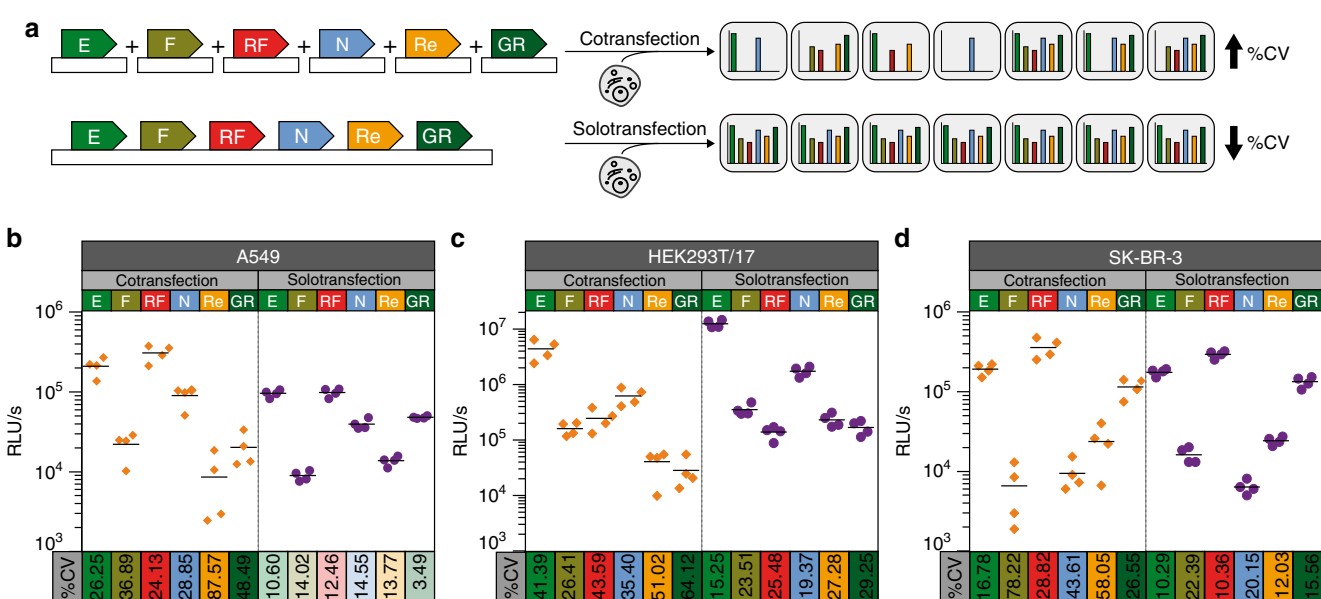

**Fig. 4 Solotransfection of a single multi-luciferase reporter vector results in lower experimental variability than cotransfection of six individual luciferase reporters. a** Schematic illustrating cellular uptake and variability issues likely encountered during cotransfection of six individual plasmids each encoding a single luciferase transcriptional unit (Top). These effects are not observed with solotransfection of a single plasmid encoding all six luciferase transcriptional units (Bottom). Equivalent cellular uptake of all luciferase units and a lower coefficient of variation (%CV) were observed with solotransfection, but not with cotransfection. **b**–**d** Variability in the quantification of the different luciferases following cotransfection (Left) or solotransfection (Right). The absolute luminescence in relative luminescence units per second (RLU/s) of four biological replicates, measured as previously described (Fig. 1i and Supplementary Fig. 7), is represented on the y-axis, while the %CV between replicates is indicated on the x-axis for A549 **b**, HEK293T/17 **c**, and SK-BR-3 **d** cells. A lower %CV was observed during solotransfection, compared to cotransfection, for all luciferase measurements in all three cell lines. E (ELuc), F (FLuc), RF (RedF), N (NLuc), Re (Renilla), GR (GrRenilla). Four technical replicates are included in each data point; the mean is represented with the horizontal bar. Source data are provided as a Source Data file.

did not observe significant changes in the TGF-β pathway (Fig. 5b, left). These results are consistent with data generated from qPCR analysis of downstream genes associated with these pathways: *CDKN1A*, *BAX1*, *IL6*, *BCL-X*, *CCL2*, and *MMP1* were down-regulated, while *E2F1* and *TERT* were upregulated. Only *VEGFD* did not exhibit any significant changes in transcript level following siRNA treatment (Fig. 5b, right). These results were corroborated when we assayed *TP53* silencing using vectors that include just one pathway reporter at a time (see Supplementary Fig. 16b, c).

On the other hand, knockdown of *SMAD2* transcript levels resulted in a 1.5-fold reduction of TGF-β pathway activity, but

only when they were previously stimulated with recombinant TGB-β protein to stimulate a pathway that demonstrates otherwise basal activity levels (Fig. 5c, Left, Supplementary Fig. 16d). Under these conditions, we also detected significant collateral downregulation of the c-Myc and upregulation of the MAPK/JNK pathway activities. These results correlate with the data obtained by qPCR of downstream genes associated with these pathways: *SMAD7*, *E2F1*, *TERT*, *MMP1* and *VEGFD* (Fig. 5c, Right).

Furthermore, according to the multiplex luciferase assay, combined knockdown of *RELA/p65* and *NFKB1/p50* transcript

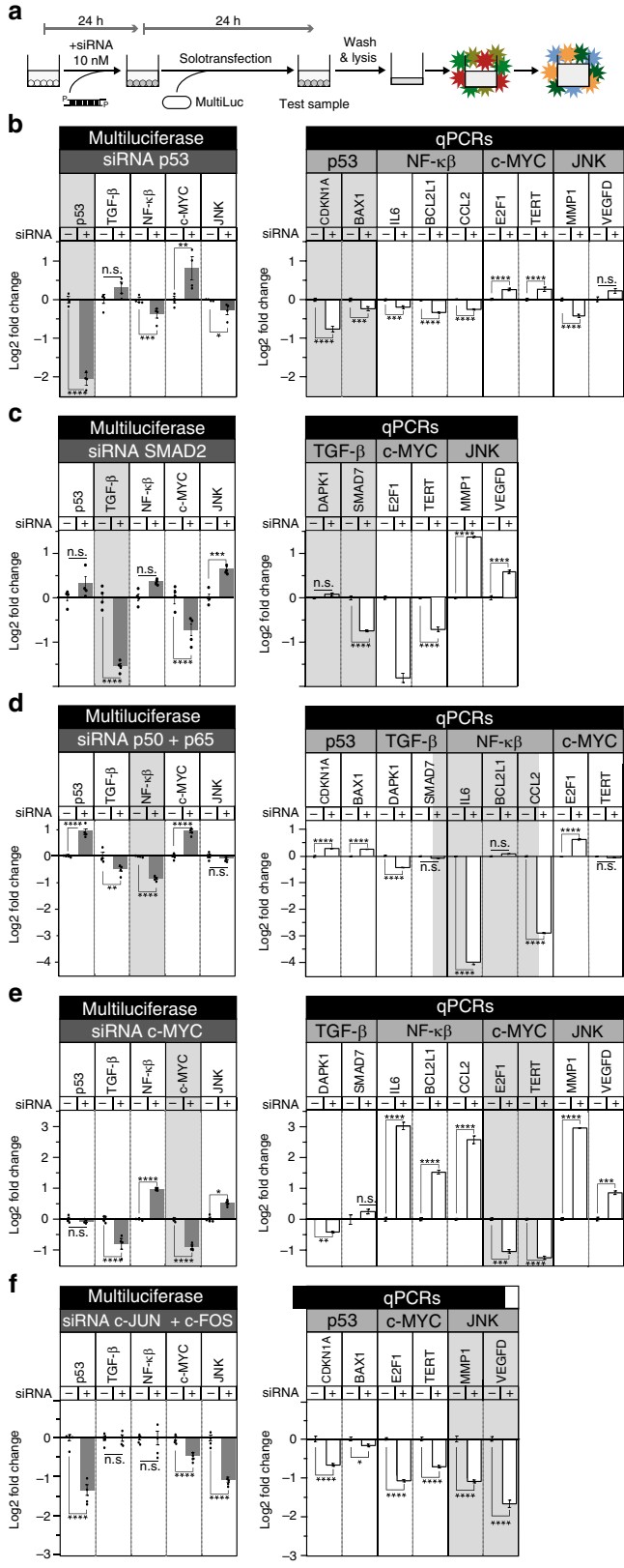

**Fig. 5 Multiplex luciferase assaying simultaneously detects the collateral and direct effects of siRNA knockdown of a single pathway. a** A549 cells were treated with 10 nM siRNA and incubated for 24 h before solotransfection of the multi-luciferase reporter. After another 24 h, cells were lysed and then the multiplex hextuple luciferase assay and quantitative PCR were performed as described. **b** The siRNA silencing of *TP53* effectively reduced p53_RE-regulated luciferase expression and decreased mRNA levels of two TP53 downstream genes, *CDKN1A* and *BAX1*. The multi-luciferase assay revealed collateral downregulation of NF-κβ and MAPK/JNK, as well as upregulation of c-Myc pathway activity, correlating with changes in mRNA levels of downstream target genes associated with each of these pathways. **c** The siRNA knockdown of SMAD2 in A549 cells previously stimulated with recombinant TGF-β protein reduced the SMAD2-regulated luciferase expression, as well as knocked down the mRNA expression levels of *SMAD7*, a key downstream target gene of this pathway. Collateral effects observed with this treatment were the downregulation of the c-Myc and upregulation of the MAPK/JNK signaling pathways, which correlated with data on mRNA expression levels of downstream genes obtained by qPCR. **d** Downregulation of the NF-κβ pathway through the simultaneous addition of siRNAs targeting *NFKB1* and *RELA* decreased the level of *IL6* and *CCL2* mRNA. Collateral effects observed with this treatment were upregulation of p53 and c-Myc, as well as downregulation of TGF-β pathway activity, that correlated with changes in mRNA expression levels of downstream target genes modulated by these pathways. **e** siRNA knockdown of the c-Myc pathway resulted in the reduction of *E2F1* and *TERT* mRNA. Collateral effects observed in this experiment were downregulation of TGF-β, and upregulation of NF-κβ and MAPK/JNK pathway activity, that correlated with changes in mRNA expression levels of several downstream target genes. **f** Interfering with the MAPK/JNK pathway through concurrent addition of siRNAs against *JUN* and *FOS* resulted in the reduction of MAPK/JNK pathway activity levels, as well as reduced MMP1 mRNA. Collateral effects observed with this treatment were the simultaneous downregulation of p53 and c-Myc pathway activity that correlated with changes in mRNA expression levels of several downstream target genes regulated by each of these pathways. Statistical significance of the fold-change of different genes analyzed by pathways in the multiplex luciferase assay and qPCR was determined by multiple *t*-tests using the Holm–Sidak method with alpha = 0.05 (*P < 0.05, **P < 0.01, ***P < 0.001, and ****P < 0.0001, n.s. is non-significant). n = 4 for both multiplex luciferase assays and qPCR experiments. Source data are provided as a Source Data file.

Supplementary Fig. 16e). These results are similar to those obtained by qPCR analysis of downstream genes. In particular, *IL6*, *CCL2*, and *DAPK1* were downregulated, while *CDKN1A*, *BAX1*, and *E2F1* were upregulated. Exceptions were *SMAD7* and *TERT* and expression of these transcripts were not altered significantly (Fig. 5d, right).

Downregulation of *MYC* transcript levels resulted in a 2-fold reduction of c-Myc pathway activity, as well as collateral repression of the TGF-β and upregulation of the NF-κβ and MAPK/JNK pathway activities (Fig. 5e, left and Supplementary Fig. 16f). A similar outcome was demonstrated by qPCR: *E2F1*, *TERT*, and *DAPK1* were downregulated, while *IL6*, *BCL-X*, *CCL2*, *MMP1*, and *VEGFD* were upregulated. Transcript levels of *SMAD7* were not significantly changed (Fig. 5e, right).

Finally, silencing of the AP1 complex using combined siRNAs against *JUN* and *FOS* resulted in a 1.1-fold reduction of MAPK/JNK pathway activity, accompanied by strong downregulation of p53 and c-Myc pathway activities (1.3- and 0.5-fold, respectively) (Fig. 5f, left and Supplementary Fig. 16g). Similar effects were demonstrated by qPCR as *MMP1*, *CDKN1A*, *BAX1*, *E2F1*, *TERT*, *MMP1* and *VEGFD* were all found to be downregulated. (Fig. 5f, right).

levels resulted in a 2-fold reduction in NF-κβ pathway activity, as well as collateral repression of TGF-β and activation of the p53 and c-Myc pathways. We did not detect significant change in pathway activity for the MAPK/JNK pathway (Fig. 5d, left and

**Collateral effects exposed after small molecule treatments**. To further confirm the utility of this approach, we analyzed the effects of pharmaceuticals (Nutlin-3 and Chetomin) and ligands (TGF-β) that perturb two of the signaling pathways included in the multiplex luciferase vector. Nutlin-3 is a small molecule that selectively activates the p53 pathway by blocking MDM2 repression of TP53. Cells with wild-type (WT) TP53 function (*e.g.*, ZR-75-1 and MCF7 cell lines) exhibit TP53 activation after Nutlin-3 treatment, followed by cell cycle arrest, apoptosis, and senescence[35] (Fig. 6a, left). In contrast, TP53 cannot be activated in cell lines that carry loss-of-function *TP53* mutations (*e.g.*, R280K mutation in MDA-MB-231 cells) or null mutations (*i.e.*, MDA-MB-157). (Fig. 6a, right). Adding Nutlin-3 during the transfection procedure results in potent, dose-dependent activation of the p53 pathway in ZR-75-1 (Fig. 6b) and MCF7 (Fig. 6c) cells, which is consistent with results produced by qPCR analysis of *BAX1* and *CDKN1A* (Supplementary Fig. 17a, b). Examples of collateral effects observed after Nutlin-3 treatment include c-Myc pathway repression and MAPK/JNK pathway activation in both ZR-75-1 and MCF-7 cells. Strong TGF-β or NF-κB repression was observed only in MCF-7 or ZR-75-1 cells, respectively. There is agreement between luciferase measurements and qPCR results for most pathways and downstream genes. Only the NF-κB pathway exhibited contradictory results in response to Nutlin-3; luciferase measurements indicated downregulation, whereas qPCR results indicated upregulation of the downstream genes for the ZR-75-1 line. (Supplementary Fig. 17a, b). As expected, p53 pathway activation was not observed after Nutlin-3 treatment of MDA-MB-231 (Fig. 6d) or MDA-MB-157 (Fig. 6e) cells, either by luciferase reporters or by qPCR analysis of genes regulated by TP53 (Supplementary Fig. 17c, d). An example of a clear collateral effect observed in both cell lines is the repression of TGF-β pathway activity. Previously, it was reported that the chemical Pifithrin-α, a well-known TP53 inhibitor, inhibits collaterally the activity of FLuc, both in vitro and in vivo, emphasizing caution when effects on reporter gene expression are being investigated, after the addition of any chemical compound[36]. Hence, to exclude the possibility that Nutlin-3 treatment has any effect on one or more of the luciferase activities, we assayed the drug against each constitutively expressed luciferase. As shown, Nutlin-3 treatment had no significant effect on any of the luciferase activities, while Pifithrin-α solely inhibited the activity of FLuc across three cell lines (Supplementary Fig. 18).

Interestingly, a small molecule called Chetomin can rescue p53 pathway activation in some *TP53* mutant cells lines[37]. SK-BR-3 cells, which carry a homozygous *TP53* point mutation (R175H), do not show p53 pathway activation upon Nutlin-3 treatment, similar to other *TP53* mutant or null cell lines. However, in the presence of Chetomin, the TP53$^{R175H}$ protein gets reactivated and restores a WT-like function (Fig. 6f). As shown, adding 10 μM Nutlin-3 alone to SK-BR-3 cells did not alter p53 pathway activity, but repressed the TGF-β pathway and activated the c-Myc and MAPK/JNK pathways (Fig. 6g). Adding 150 nM Chetomin alone resulted in several changes: p53 pathway became activated, c-Myc pathway became more activated, NF-κB and MAPK/JNK pathways were repressed, and TGF-β pathway was no longer repressed (Fig. 6g). The addition of 10 μM Nutlin-3 with 150 nM Chetomin produced an outcome that was similar to adding Chetomin alone, except that p53 pathway activity was further enhanced, and NF-κB pathway activity was further repressed. The MAPK/JNK pathway repression was neutralized (Fig. 6g). The luciferase measurements were confirmed by qPCR analysis of downstream genes (Supplementary Fig. 19), while the non-specific inhibition effects of Chetomin on the six luciferase activities was excluded by assaying the drug against each constitutively-expressed luciferase (Supplementary Fig. 18).

Finally, we performed the multiplex luciferase assay to examine the effects of recombinant TGF-β on TGF-β-responsive (MDA-MB-231 and MCF7) or insensitive (ZR-75-1 and SK-BR-3) breast cancer lines (Fig. 6h). Adding 5 ng/mL of recombinant TGF-β at 16 h post-transfection resulted in different responses among the four cell lines, 6 h after ligand addition. MDA-MB-231 and MCF7 cells demonstrated significant activation of TGF-β pathway activity (Fig. 6i, j), while ZR-75-1 and SK-BR-3 cells were unaffected (Fig. 6k, l). An additional collateral effect, namely slight downregulation of the c-Myc pathway, was observed in MDA-MB-231 cells (Fig. 6i). All of these observations were confirmed by qPCR analysis of downstream genes with the exception of *CDKN1A*, *IL6*, and *CCL2*, which were upregulated (Supplementary Fig. 20a-d). These data are consistent with previous findings in other cell lines that demonstrated the presence of SMAD-binding elements within the *CDKN1A*[32] and *CCL2*[38] promoters or direct crosstalk between the TGF-β and NF-κB signaling pathways from the *IL6* promoter[39,40]. Moreover, recombinant TGF-β did not demonstrate any significant effect on one or more of the luciferase activities by assaying the drug against each constitutively expressed luciferase (Supplementary Fig. 18).

## Discussion

Dual-luciferase assays are widely employed throughout biomedical research fields[12,13]. Despite plentiful advances in the luminescence field[13], the number of luciferases that can be detected simultaneously in a single experiment has remained limited. One innovation toward multiplexing is the implementation of two luciferases that produce distinguishable emission spectra, using a single[41] or separate, orthogonally acting[14] substrates. Another innovation incorporates the use of two luciferases with emissions that can be deduced from partially overlapping substrate usage, *i.e.*, one substrate is used by both luciferases and a second substrate is used by only one[42]. More advanced innovations toward multiplexing allow for the simultaneous detection of three luciferases with activities that can be spectrally distinguished using appropriate emission filters after the addition of one[18] or more substrates[43]. Alternatively, three luciferases that each use a unique substrate have been used to monitor three distinct cellular phenomena in vitro[44] or, sequentially, in vivo[45]. Thus, currently the maximum number of luciferases that can be measured simultaneously is only three. Since applications of these tri-luciferase detection methods have not been broadly implemented, it is fair to say that today luciferase multiplexing is limited to two.

Here we demonstrate an approach that enables the detection of six luciferases simultaneously in a single endpoint post-lysis experiment using standard, well characterized, reagents. Our hextuple reporter assaying method multiplexes six luciferase enzymes whose activities can be uniquely determined by combining orthogonal substrate usage, selective quenching of the luminescence, and spectral decomposition. The assay enables the examination of five cellular activities against a constitutively active reporter. Successful implementation of the assay requires that appropriate filters and readout times are used, and that luminescences are within the dynamic range of the assay, which spans from >$10^6$ to ≥$10^4$ RLU/s for the D-Luciferin luciferases and from >$10^7$ to ≥$10^5$ RLU/s for the coelenterazine luciferases. To facilitate this assaying, we also established a versatile and adaptable cloning pipeline to generate a multigenic vector that contains all six luciferase reporter units. While we incorporated reporter elements for five specific cellular pathways, any other pathway could easily be included depending on the biomedical problem to be probed. Moreover, this work demonstrates that solotransfection of a single multi-reporter is favored over

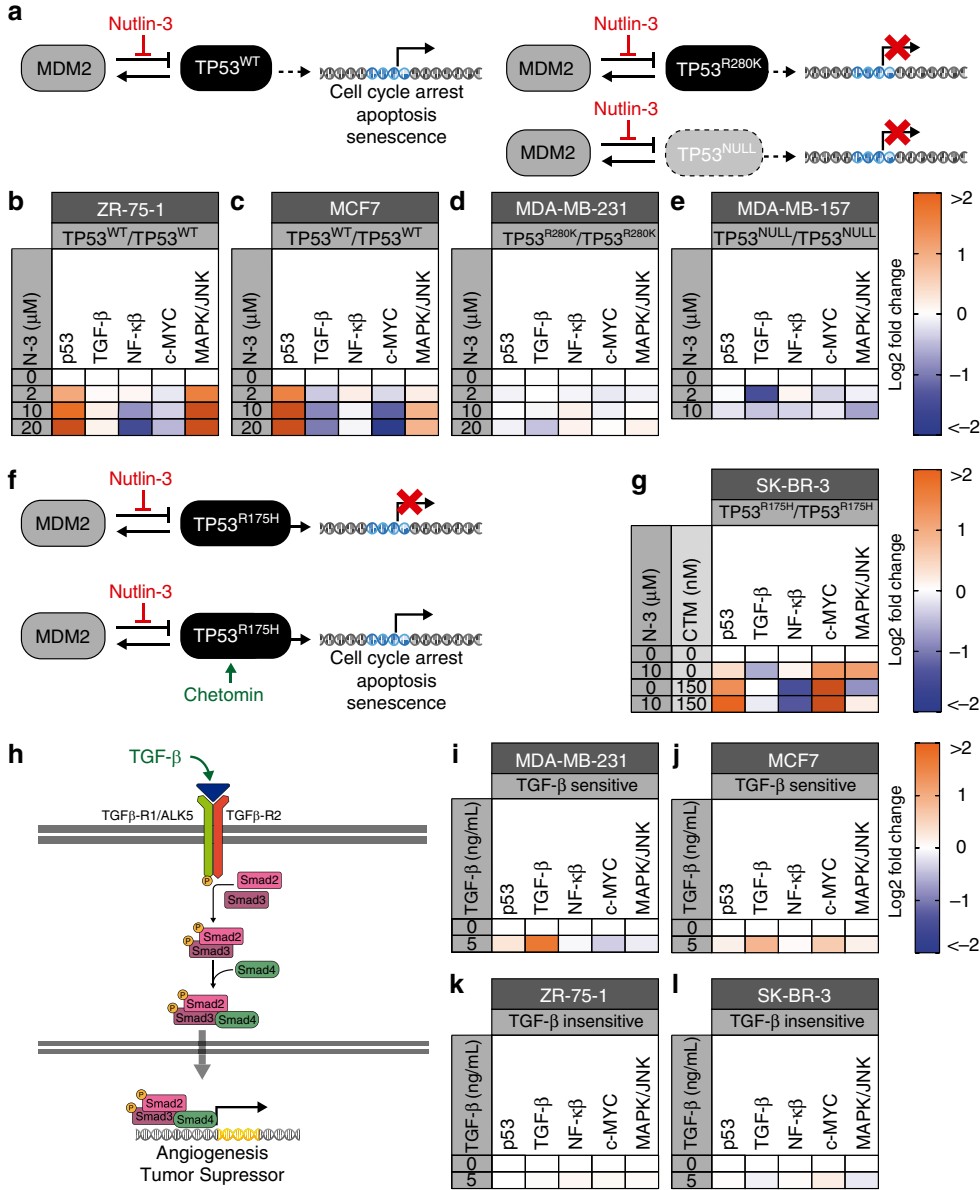

**Fig. 6 Multiplex luciferase assaying simultaneously detects direct and collateral effects in breast cancer cell lines after treatment with pathway-specific pharmaceuticals or ligands. a** Schematic of p53 pathway activation induced by Nutlin-3. Nutlin-3 selectively inhibits the MDM2-TP53 interaction and activates the p53 pathway in a cell line that expresses wild-type TP53 (TP53[WT]). Cell lines that express mutant TP53 (TP53[R280K]) or are p53-deficient (TP53[NULL]) are resistant to Nutlin-3-induced activation of the p53 pathway. **b–e** Heat maps showing the effect of 24-hour treatment with increasing concentrations of Nutlin-3 (N-3) in four breast cancer cell lines: ZR-75-1 **b** and MCF7 **c** are WT, while MDA-MB-231 is mutant **d**, and MDA-MB-157 is null **e**. The p53 pathway is only activated in the two TP53[WT] cell lines. Collateral and differential effects across the other pathways examined are shown. **f** Schematic of MDM2-p53 pathway reactivation by the pharmaceutical Chetomin. Cell lines that express R175H TP53 mutant protein (TP53[R175H]) are resistant to Nutlin-3-mediated inhibition of MDM2 and activation of the p53 pathway. In the presence of Chetomin, TP53[R175H] is reactivated and functionally restored to WT-like levels. **g** Heat map demonstrating the complementary effect of Nutlin-3 and Chetomin on p53 signaling. SK-BR-3 cells were not significantly affected by Nutlin-3. The addition of 150 nM of Chetomin reactivates the p53 pathway in the presence and absence of Nutlin-3. Collateral and differential effects on the other pathways are shown for all treatments. **h** Schematic of TGF-β pathway activation by TGF-β ligand. TGF-β activates cellular signaling through the heterodimeric receptor TGF-β-R1/ALK5 and TGF-β-R2, resulting in transcriptional activation of downstream genes mediated by Smad2/Smad3/Smad4. **i–l** Effect of a 6-h treatment with 5 ng/mL of recombinant TGF-β in four breast cancer cell lines: MDA-MB-231 **i**, MCF7 **j**, ZR-75-1 **k**, and SK-BR-3 **l**. Addition of the ligand resulted in significant activation of the downstream luciferase reporter in TGF-β-sensitive lines (MDA-MB-231 and MCF7). Significant downregulation of the c-Myc pathway was also observed in these cell lines. The addition of the ligand did not result in significant pathway activation in TGF-β-insensitive cell lines (ZR-75-1 and SK-BR-3). $n = 4$ for all multiplex luciferase experiments. Source data are provided as a Source Data file.

cotransfection of six individual vectors, decreasing variability between biological replicates and providing an additional level of experimental control. In conclusion, our approach reported here allows for simultaneous readout of transcriptional activity of five cellular activities providing a much deeper understanding of cellular pathway activities of interest.

The development of a technology or the substantial expansion of an existing technology always raises concerns and limitations. One concern of our method may be the size of the plasmid used (13.4 kb), potentially limiting its applicability using hard to transform cell lines. However, Lipofectamine 3000-mediated transfection, the transformation method used in the present study, can be substituted for by any other method, including many other lipofection or chemical transfection methods, as well as electroporation, sonoporation, and viral transduction methods[46]. Alternatively, stable cell line clones can be isolated by selection after any of the above mentioned transformation methods, previously demonstrated to work for plasmids up to hundreds of kilobases in size, including bacterial artificial chromosomes[47–49]. To reduce variability of measurements as much as possible, stable cell lines could be generated by the targeted integration of all multiplex reporters at the same defined site in the genome, such as a safe harbor site[50], resulting in the neutralization of genomic position effects. Cells with such genome-integrated luciferase reporters can then be further used to explore luciferase multiplexing towards in vivo bioluminescence applications[51–53]. Another concern may be potential cross-talk between the reporters integrated into the same plasmid by synthetic biology. In our multiplex reporter construct, besides a transcription terminator downstream of each transcriptional reporter unit, we have also included a well-characterized transcription blocker element in between each of the units (i.e., a synthetic polyA terminator and the RNA polymerase II transcriptional pause signal from the human α2 globin gene), to prevent transcriptional interference between different pathway-responsive luciferase transcriptional units[31]. In this work, we demonstrate that cross-talk between the different response elements located on the same plasmid couldn't be observed. Also, experiments should exclude the possibility that the tested drugs may inhibit the enzymatic activities of one or more luciferases in the assay, as previously reported for Pifithrin-α[36]. This becomes especially important in high-throughput drug screens, where numerous candidates will be identified of which some may be false positive hits due to interference with luciferase activity[36,54,55], careful follow-up studies, as illustrated in this work, should be performed to exclude interference of isolated compounds on luciferase activity. Furthermore, while the current study involves intracellularly localized luciferases, endpoint post-lysis experimentation only allows for a single multiplex measurement. One potential solution is the adding of a membrane export signal to each luciferase to help their extracellular secretion[56]; combining this approach with the generation of stably integrated cell line clones, longitudinal reporter studies and multiple multiplex measurements become possible by taking small media samples for analysis at defined points over time. Finally, in this study, native luciferases without the addition of PEST proteolysis sequences[57] were used, resulting in high basal expression levels and subsequently low fold induction levels. Those induction levels could be improved by adding such PEST sequences to each luciferase.

In future work, more luciferases could be integrated to expand the capabilities of this assay. Approaches to accomplish this include molecular modification of existing luciferases into regions of emission spectra not yet covered by the D-Luciferin- or coelenterazine-responsive luciferases or investigation of luciferases that utilize other substrates such as vargulin and furimazine[13,51]. Additional luciferases can be easily incorporated in our multiplex pipeline by synthetic assembly[25–27]. It is noteworthy to mention that plate reader hardware currently only provides two injectors and therefore only accommodates the simultaneous use of two luciferase substrates. We hope that this work will stimulate plate reader hardware improvements to accommodate the implementation of three or more luciferase substrates in the near future. Alternatively, compatibility with other non-luciferase reporters could be explored to expand the assay capabilities even further.

Besides applications in cancer research, as demonstrated here, hextuple luciferase assaying can be used to study other cellular pathways or complex diseases. By using the hextuple luciferase reporter system, it will be possible to simultaneously monitor the activities of five nuclear receptors by placing the specific regulatory elements upstream of five different luciferases[58], making it possible to measure nuclear receptor cross-reactivities, determine natural ligand specificity, and screen for agonists and antagonists at the same time. Hextuple luciferase assaying could also be used to monitor five known transcription factor activities downstream of the insulin receptor signaling pathway[59], enabling an investigator to probe environmental factors and pharmaceuticals affecting insulin sensitivity in different cell types. The assay could be tailored to study the innate immune responses of host cells during viral infection, which is predominantly mediated by three types of receptors[60]. Furthermore, measuring the activities of transcription factors acting downstream of these pathways would reveal correlations between viral susceptibility and the innate immune response. Multiplex luciferase assays will also be tremendously helpful in the quantification and comparison of genetic regions for synthetic biology applications both in vitro and, with some adaptations in vivo. Current methods to assess these regions and enable predictive engineering of synthetic genetic circuits rely mostly on dual-luciferase assay outputs[61]; therefore, implementing multiplex luciferase assay approaches will increase the rate and reliability of these assessments. Finally, while luciferase multiplexing was solely explored in the context of in vitro transcriptional reporter assaying during this study, similar approaches can be implemented to explore multiplex capabilities for other in vitro approaches (protein translation reporters, biosensors and others), as well as in vivo bioluminescence imaging[51].

In conclusion, the dual-luciferase reporter system has become a staple of modern biological research and the necessary hardware has become reasonably ubiquitous. Multiplexed versions will likely be adopted across a diversity of biological disciplines, and the need for more information-rich experimental designs will lead to applications beyond the traditional dual-luciferase system.

## Methods

**DNA synthesis, synthetic assembly, and molecular biology**. Molecular biology experiments, including plasmid maps and *in silico* experimentation, were designed and generated using SnapGene software (http://www.snapgene.com/products/snapgene/) (GSL Biotech LLC). Synthetic DNA assemblies were performed using the GoldenBraid 2.0 DNA assembly framework, a Type IIs restriction enzyme cloning method based on the use of two levels of plasmids (Alpha and Omega) that can be combined in successive rounds of assemblies (going from Alpha to Omega, back to Alpha to Omega, and so on) to create a perpetual loop allowing for virtually indefinite growth of assemblies (limited only by vector backbone properties and the stability of insert DNA). GoldenBraid 2.0 allows a convenient way to combine multiple transcriptional units into a single DNA strand[25–27]. Synthetic DNA was obtained as oligonucleotides (Sigma-Aldrich), DNA fragments or blocks (IDT and Eurofins Genomics), or cloned DNA (Twist Bioscience). All DNA blocks were transferred to the domestication vector pUPD[25,26], or a domestication vector generated for this project (pUPD3) (Supplementary Fig. 21a, b). All domestication experiments were confirmed by agarose gel DNA electrophoresis after restriction enzyme digestion to expose diagnostic DNA bands of specific lengths (restriction enzymes were purchased from New England BioLabs), as well as control uncut plasmid to eliminate unwanted multimeric assemblies. In addition, we built high-

copy vectors with a ColE1 origin of replication (pColE1_Alpha1, pColE1_Alpha2, pColE1_Omega1, and pColE1_Omega2), allowing for GoldenBraid 2.0-based stitching of insert assemblies up to about 25 kb (Supplementary Table 4). The multi-luciferase reporter plasmid was built with successive rounds of assembly (Fig. 3), and included five luciferases under the control of pathway-specific transcriptional response elements (Supplementary Table 4) and a sixth luciferase (ELuc) that served as the internal standard for normalization purposes. All assembly steps were performed in one-pot-one-step reactions (Fig. 3c–e, Supplementary Fig. 10b-d, and Supplementary Fig. 13)[25,26]. To summarize, 75 ng of the destination vector and 75 ng of each of the parts to be assembled were mixed with 1 µL of the appropriate restriction enzyme (BsaI or BsmBI, New England BioLabs), 1 µL of T4 Ligase, and 1 µL of the Ligase 10× Buffer (Promega) in a final volume of 10 µL. Reactions were set up in a thermocycler with 25 cycles of digestion/ligation reactions (2′ at 37 °C, 5′ at 16 °C). In toal 2 µL of the reaction were transformed into DH10β chemocompetent *E. coli* cells (Thermo Fisher Scientific) and positive clones were selected on solid media containing X-Gal and the appropriate antibiotic: 100 µg mL$^{-1}$ ampicillin for GBParts assembled in pUPD, 30 µg mL$^{-1}$ kanamycin for Alpha destination vectors, and 12.5 µg mL$^{-1}$ chloramphenicol for the GBParts assembled in pUPD3 and Omega destination vectors. Only white colonies were pursued further; blue colonies are reconstituted destination vectors. Plasmid DNA from white colonies was extracted using the ChargeSwitch-Pro Plasmid Miniprep Kit (Thermo Fisher Scientific). Assemblies were confirmed by agarose gel DNA electrophoresis after restriction enzyme digestion to expose diagnostic DNA bands of specific lengths (restriction enzymes were purchased from New England BioLabs), as well as control uncut plasmid to eliminate unwanted multimeric assemblies (Fig. 3g and Supplementary Fig. 14). All empty backbone vectors, initial domesticated parts, constitutively expressed luciferase units, and the final multi-luciferase reporter plasmid are available through Addgene (https://www.addgene.org/) (Supplementary Table 4). All plasmid maps can be viewed and analyzed using SnapGene software (above) or SnapGene Viewer freeware software (http://www.snapgene.com/products/snapgene_viewer/) (GSL Biotech LLC). All primers used in this work are listed in Supplementary Table 8. Synthetic DNA fragments are listed in Supplementary Table 9.

**Cell culture**. All cell lines used in this study were obtained from the Tissue and Cell Culture Core at Baylor College of Medicine (MCF7/ATCC HTB-22, MDA-MB-231/ATCC HTB-26, SK-BR-3/ATCC HTB-30, ZR-75-1/ATCC CRL-1500, A549/ATCC CCL-185, HEK 293T/17/ATCC CRL-11268), and the Characterized Cell Line Core Facility at MD Anderson (MDA-MB-157/ATCC HTB-24). Cells were cultured according to standard mammalian tissue culture protocols and sterile techniques (Supplementary Table 10). Cell lines were incubated at 37 °C and 5% $CO_2$. Specific growth media, cellular density, and plate format used for siRNA and luciferase reporter plasmid transfections are summarized for each cell line in Supplementary Table 10. For quantitative PCR, cells were plated at a density of 4 × 10$^5$ cells/well in 6-well plates for RNA extraction. To confirm the identity of all cell lines used in this study, short tandem repeat DNA fingerprinting was performed at the Characterized Cell Line Core Facility (MD Anderson) (Supplementary Table 11).

**Plasmid and siRNA transfection**. Lipofectamine 3000 (Thermo Fisher Scientific) was used for all plasmid transfections. We used the standard transfection protocol indicated in the accompanied manual for plasmids smaller than five kilobases, in a 96-well plate: 100 ng plasmid, 15 µL Opti-MEM (Thermo Fisher Scientific), 0.2 µL P3000 reagent, and 0.2 µL lipofectamine reagent was used in a total volume of 15.4 µL per sample (with addition of the volume of the DNA) to prepare the DNA-plasmid complexes. We used a modified transfection protocol for plasmids larger than five kilobases: 150 ng plasmid DNA, 15 µL Opti-MEM (Gibco), 0.3 µL P3000 reagent, and 0.3 µL lipofectamine reagent in a total volume of 15.6 µL per sample (with addition of the volume of the DNA) was used to prepare the DNA–plasmid complexes. When 48-well plates were used, volumes were adjusted proportionally: twice the amounts of all reagents were used per sample to prepare the DNA-plasmid complexes.

Lipofectamine RNAiMAX Reagent (Thermo Fisher Scientific) was used for all siRNA transfections as recommended by the manufacturer. All siRNAs (Supplementary Table 6) were purchased from Sigma-Aldrich and transfected at a final concentration of 10 nM. The MISSION® siRNA Universal Negative Control #1 (Sigma-Aldrich) was used as non-targeting siRNA to create a baseline for mRNA knockdown efficiency, and the knockdown potency of all targeting siRNAs were normalized to this negative control.

**Measurement of luciferase activities and bioluminescent spectra**. Luciferase activities were measured using the CLARIOStar multimode microplate reader (BMG LABTECH GmbH), incorporating appropriate bandpass (BP) filters when necessary. All luciferase emission measurements were performed using the Dual-Luciferase® Reporter assay (DLR™ assay, Promega). At 24 h post-transfection (with or without treatments as indicated), cells were washed with 150 µL phosphate buffered saline (PBS) per well for 96- or 48-well plates, followed by cell lysis using 40 or 60 µL of Passive Lysis Buffer (PLB) per well for 96- or 48-well plates,

respectively. The necessary amount of lysate required for experiments (below) were transferred to white 384-well plates for luminescence recordings.

We recorded full-spectrum luminescence for individual luciferase measurement using the emission Linear Variable Filter (LVF) monochromator of the CLARIOStar multimode microplate reader: bandwidth was set to 10 nm and measurements were taken from 350 to 700 nm in 1 nm steps. Luciferase measurements were initiated 5 s after addition of 10 µL of the appropriate buffer (containing appropriate substrate) to 5 µL of the cell lysates: LARII buffer (containing D-Luciferin) alone for the D-Luciferin-responsive luciferases, or followed by the addition of Stop & Glo® Buffer containing quencher and coelenterazine for the coelenterazine-responsive luciferases. The performance of all luciferases was also evaluated by just adding buffer containing quencher and coelenterazine, without previously adding D-Luciferin-containing buffer. This resulted in luminescence readings lower than the background threshold set for the D-Luciferin luciferases and weaker luminescence readings for the coelenterazine luciferases. To improve the signal-to-noise ratio, the smoothing algorithm of the MARS Data Analysis Software (BMG Labtech) was applied with a boxcar width set to 9 for all samples. Subsequently, we normalized the data by setting the maximum emission peak at 100%.

For multiplex luciferase recordings, we determined the D-Luciferin luminescence 30 s after the addition of 10 µL of LARII buffer (containing D-Luciferin substrate) by measuring (1) total, (2) BP515-30-filtered, and (3) BP530-40-filtered light for 2 s each. In a second step, we determined the coelentarazine luminescence 7 s after adding 15 µL of Stop & Glo® buffer (containing quencher to annihilate the luminescence from the first step, and coelenterazine substrate) by measuring (1) BP410-80-filtered, (2) BP570-100-filtered, and (3) total light for 1 s each (see Supplementary Fig. 7 for a schematic of the protocol).

Two additional luciferase assay kits, Nano-Glo® luciferase assay (Promega) and Dual-Glo® luciferase assay (Promega), were tested in this study but ultimately not used for the multi-luciferase experiments.

**Total protein quantification**. The Pierce BCA Protein Assay Kit (Thermo Fisher Scientific) was used to measure the total protein content in cell lysis samples. Reagents were prepared as indicated in the manual, and 1 µL of the cell lysate was mixed with 40 µL of the reagent in a 384 well plate. Absorbance was measured at 562 nm in a CLARIOStar multimode microplate reader, and total protein concentration was determined by comparing to a bovine serum albumin protein standard. Three technical replicates were performed per protein quantification experiment.

**RNA extraction, cDNA synthesis, and quantitative PCR (qPCR)**. RNA extraction was performed after drug or siRNA treatment for 24 or 48 h, respectively. Cells were quickly trypsinized, harvested by centrifugation, and pellets were frozen until further processing. RNA was isolated using the Quick-RNA™ MicroPrep kit (Zymo Research) as recommended by the manufacturer, including the optional DNAse treatment. First-strand cDNA synthesis was performed in a 20 µL reaction using 1000 ng of RNA and 4 µL of the qScript cDNA SuperMix (Quantabio). Next, 4 µL of a 1:5 dilution of cDNA was used for qPCR, combined with 5 µL of PerfeCTa SYBR® Green SuperMix (Quantabio), and 1 µL of a 55 µM primer mix (containing both forward and reverse PCR primers) per sample. Quantitative PCR was performed in 384-well qPCR plates (Roche) using the LightCycler® 480 Real-Time PCR instrument (Roche). qPCR data analysis was performed using qbase +3.0 software (Biogazelle)[62]. Primers used for qPCR (candidate reference genes and pathway genes) are listed in Supplementary Table 12. We used the geNORM algorithm[63] to determine expression stability of candidate reference genes and the optimal number of reference genes (Supplementary Fig. 22a, b).

**Data analysis and statistics**. All data were analyzed using four worksheets of an Excel file provided as Supplementary Data 1: worksheet 1 provides formulas for the calculation of transmission coefficients (as explained in Supplementary Fig. 23), worksheet 2 provides formulas for the calculation of simultaneous equations (as explained in Supplementary Fig. 24), worksheet 3 provides formulas for the unformatted measurements from a small group of samples (as explained in Supplementary Fig. 25), and worksheet 4 provides formulas for the Unformatted measurements from a large group of samples: 96-well format (as explained in Supplementary Fig. 26). Analyzed data were then migrated into Prism 7 software (GraphPad) for statistical analysis and graphing. The resulting graphs were then edited for publication using Adobe Illustrator CC (Adobe Creative Cloud). For linear regressions, Prism reports the P-value test from the null hypothesis that the overall slope is zero, and it is calculated from an F test. For qPCR and multiplex luciferase assays, the log$_2$ fold-change was calculated (see Supplementary Table 7 for comparison with percent change between samples and fold-change). Statistical significance of the fold-change in qPCR and multiplex luciferase assays was determined by the multiple t-test using the Holm–Sidak method with alpha = 0.05 (*$P < 0.05$, **$P < 0.01$, ***$P < 0.001$, and ****$P < 0.0001$, n.s. is non significant).

**Reporting summary**. Further information on research design is available in the Nature Research Reporting Summary linked to this article.

## Data availability

The source data underlying Figs. 1b-h, 2b-d, f-h, 4b-d, 5b-f, 6b-e, g, i-l, Supplementary Fig. 5b-e, 6b-e, 8b-d, 9b-d, 11b-d, 12b-d, 15a-e, 16a, c-g, 17a-d, 18b-d, 19, 20a-d, and Supplementary Tables 1 and 2 are provided as a Source Data file. Original DNA agarose gels are included in the Source Data file.

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

## Acknowledgements

We thank Cheryl Parker and the Tissue and Cell Culture Core at Baylor College of Medicine for technical advice related to mammalian cell culture work, Herman Dierick, Ido Golding, Christophe Herman, and Anna Sokac for comments, Kevin MacKenzie for editing, Nick Matinyan for the pUPD3 vector, and Eric Matthews (BMG LABTECH) for technical assistance with the CLARIOStar luminometer. pUPD[26] and pLuciferase (Addgene #68201) vectors were gifts from Diego Orzaez (Institute for Plant Molecular and Cell Biology—IBMCP). This work was supported by start-up funds kindly provided by Baylor College of Medicine (D.W.Y. and K.J.T.V.), the Albert and Margaret Alkek Foundation (K.J.T.V.), the McNair Medical Institute at The Robert and Janice McNair Foundation (K.J.T.V.), as well as a March of Dimes Foundation grant #1-FY14-315 (K.J.T.V.), the Foundation For Angelman Syndrome Therapeutics grant FT2016-002 (K.J.T.V.), the Cancer Prevention and Research Institute of Texas grants R1313 (K.J.T.V.) and R1314 (D.W.Y.), and the National Institutes of Health grants 1R21GM110190 (K.J.T.V.), 1R21OD022981 (K.J.T.V.), R01GM109938 (K.J.T.V.). The Dan L. Duncan Comprehensive Cancer Center is supported by Cancer Center Support Grant P30 CA125123 (National Institutes of Health and National Cancer Institute). Simple tandem repeat DNA fingerprinting was performed by the MD Anderson Characterized Cell Line Core Facility, supported by Cancer Center Support Grant P30 CA016672 (National Institutes of Health and National Cancer Institute). Plasmids are available through Addgene (https://www.addgene.org/): accession numbers are available in Supplementary Table 3.

## Author contributions

A.S.-P. and K.J.T.V. conceptualized the study. A.S.-P., L.C., D.W.Y., and K.J.T.V. planned and designed the study. A.S.-P., L.C., Y.G., and T.G.-F. performed experiments. A.S.-P., L.C., and Y.G. analyzed the data. A.S.-P and K.J.T.V. drafted the manuscript, which was revised according to feedback from all other authors. K.J.T.V. and D.W.Y. supervised the study.

## Competing interests

The authors declare no competing interests.
