## [Peer Review File · Nature Communications]

Reviewers' comments:

Reviewer #1 (Remarks to the Author):

This is a well written and a technical tour de force study that develops the technology to measure 6 luciferase reporters simultaneously in cells. The technical aspects of the study are well controlled and expertly performed.

The only major issues I have concern the specificity of the pathway reporter elements that were finally combined as a proof of principle and as a way of measuring "collateral" pathway signalling activity.

1) Firstly, when combined in a single plasmid does the collateral effects truly reflect collateral effects or are they a consequence of cross-talk on the plasmid itself? I would like to see the authors co-transfect the individual reporters and then measure the effects of siRNA knockdown of each pathway on each individual pathway.

2) Do the reporters reflect the activity of each pathway? The authors should measure the activity of the transcription factors reported in at least one other assay such as DNA- binding ability, nuclear translocation etc for each of the transcription factor complexes and assess potential collateral effects in these assays also.

3) In Figure 5 all pathway outputs should be shown in all knockdowns. There is no data shown for SMAD2 knockdown for instance. C-fos knockdown did not work and in fact reduced the effect of c-jun knockdown on c-fos levels. This should be repeated with an efficacious siRNA.

Minor points:

1) The final plasmid should be sequenced

2) Figure 1 legend needs n numbers and errors added

3) Some figures are quoted to 5 decimal places (eg sup Fig3 supp table 1,2) which will overestimate the accuracy of the measurements.

4) Supp Fig 7 remove "see e" and "see f"

Reviewer #2 (Remarks to the Author):

This manuscript describes the development of a multiplex luciferase assay system which allows simultaneous interrogation of multiple (up to 5, plus a control) cellular response pathways. The approach is novel and meticulously performed. The careful systematic presentation of the development of the assay, although lengthy, provides a useful and rich background for other investigators desiring to build upon this work and I think should be included as written (in other words, please resist shortening the manuscript by deleting these sections). As a methods paper, it is critical that the authors describe in detail all of the necessary steps needed to replicate the work, and the authors have done an excellent job conforming to this requirement. It is commendable to also include the spreadsheet for data analysis and to provide the plasmids used.

Overall this is an excellent manuscript and I think will be of interest to a widespread audience. This type of assay does compete with other existing technologies, such as multiplex qPCR and Luminex assays which are already well established and perhaps easier to set up. It is possible that this multiplex luciferase assay could be adapted for high throughput drug screening, though additional work would be required to establish its potential in that role. More likely, novel uses will be developed in the future that may be not obvious at present.

Minor Criticisms:

1). In the qPCR assays, it would be useful to include direct measure of the luciferase transcripts to compare with the enzyme activity. Only downstream targets, which are indirect readouts, are included at present.

2). The use of the language "the CMV promoter" is slang, and is more correctly referred to as the hCMV major immediate early promoter or at least, the hCMV-IE1 promoter.

3). In the paragraph of speculation of the potential applications of this method in the Discussion, it would be more useful to include a balanced description of the limitations of the method. For

example, the assay is best applied to easily transfected lines. Another limitation is the size of the plasmid used (13.4 kb), as larger plasmids become more difficult to transfect successfully. It is also possible that there could be cross-talk between reporters (for example, through enhancer action) when incorporated on the same molecule, although this does not appear to be the case under the conditions shown in the study.

Reviewer #3 (Remarks to the Author):

The authors report a really fantastic and innovative approach to quantifying the activity of multiple different bioluminescent reporters of gene expression from the same cell population. The manuscript, though data-dense, is extremely well presented and very clearly written. The authors have produced a very useful tool that has the potential to become invaluable in the delineation of cross-talk between signal transduction pathways and their downstream transcriptional effectors. I applaud their efforts.

Based on my understanding of the advance I have a single major criticism, that I think can be addressed quite simply and quickly - my sincere apologies if I have misunderstood:

... the utility of this technique rests upon the assumption that the luciferase mRNA transcribed from any given promoter sequence, following some perturbation, correlates directly with the level of bioluminescence detected in the destructive endpoint assay that follows. This is not always a safe assumption, since it is quite plausible that some genetic or pharmacological perturbations may affect the stability of one or more luciferase mRNAs or proteins or their translation rate. Moreover, several luciferase-based small molecule screens have identified compounds that eventually turned out to be very effective inhibitors or activators of luciferase activity, but irrelevant to the pathway under investigation.

To control for all of these possibilities, in the experiment shown Fig 6 in particular, I would consider it essential to repeat the experiment, with all 6 luciferases under the control of the same promoter in order to exclude the possibility that the effect of each drug treatment may instead be attributable to some post-transcriptional effect on one or more of the luciferase activities. Ideally this would occur by the same solotransfection method, but it would be acceptable to simply repeat the drug assays using each of the 6 CMV driven luciferases (Supp table 4) separately, in the wild type cell line, presented in an additional supplementary figure. In order to increase the robustness and reproducibility of that the wonderful technology they have developed, the authors might consider including a sentence in the discussion as to why it is important to control for such off-target effects in luciferase-based screens.

Minor comments

1c seems to be a replot of some of the same data presented in 1b rather than a separate experiment. If so this should be mentioned explicitly in the legend.

Page 6, line 2 & abstract - many researchers use luciferases for longitudinal, live cell assays. The manuscript describes a destructive endpoint assay, as described in other parts of the manuscript. So as not to confuse the naive reader, it might be worth reiterating at this point that the the various luciferase activities measured post-lysis. In the interests of clear communication, it might be worth also mentioning that the endpoint nature of the assay in the abstract.

Page 11, line 11 - the observation that coelenterazine-responsive luciferases are associated with high experimental error requires some discussion. I do not doubt the observation, but find it hard to rationalise, as may others.

Page 12, line 5 - the authors are of course welcome to call this 'solotransfection' if they wish, but the term is unlikely to catch on because it doesn't distinguish what the authors have done, from the fairly standard practise of transfecting cells with single plasmids that express more than one protein. I suggest that they come up with a better name if they wish it to enter common parlance.

siRNA controls - I accept that the siRNA knockdown data presented in this manuscript is showing proof-of-principle, but its very unusual these days not to include scrambled or non-targeting siRNA sequences in order to control for non-specific effects that transfection and introduction of exogenous small RNAs into the cytoplasm has on cells - this may trigger dysregulation of the endomembrane system or an innate intracellular immune response, for example, which could interfere with the assay result. So far as I can discern, no siRNA controls are reported in this manuscript. It is not sufficient to report that these sequences have been validated previously, it is important to show that transfection of each cell line of interest with some non-relevant siRNA did not affect the activity of the hextuple reporter or if effects are observed in the controls, that these be quantified and taken into account. If the authors do not wish to perform siRNA controls, they should state explicitly that no siRNA control transfections were performed in the legend of figure 5, the methods section and the relevant portion of the results.

CMV promoter - Are the authors confident that the CMV promoter is constant under all conditions. In our experience its activity can vary greatly between different cell culture conditions. In the context of a high content screen, for example, It was not quite clear to me how some exogenous perturbation that affects transcriptional regulation at the CMV promoter would be taken into account. Apologies if I have missed this.

Please find enclosed the resubmission of our manuscript “**Examining multiple cellular pathways at once using multiplex hextuple luciferase assaying**” by Alejandro Sarrion-Perdigones, Lyra Chang, Yezabel Gonzalez, Tatiana Gallego-Flores, Damian Young, and Koen Venken. Please find below our response to the comments and critiques raised by the three reviewers.

Reviewer 1

Major points:

- 1. Firstly, when combined in a single plasmid does the collateral effects truly reflect collateral effects or are they a consequence of cross-talk on the plasmid itself? I would like to see the authors co-transfect the individual reporters and then measure the effects of siRNA knockdown of each pathway on each individual pathway.**

We have performed numerous attempts in order to comply with the request of the reviewer regarding cotransfecting six single luciferase reporters and then measure the effects of siRNA knockdown of each one on each individual pathway. Unfortunately, the experimental errors due to cotransfection were very high, as shown in the new Supplemental Figure 16a. In this panel, we demonstrate how the cotransfection of 2, 3, 4, 5 or 6 vectors result in a high coefficient of variation for the light measured across the different biological replicates, compared to the overall error obtained when the multiplex hextuple luciferase reporter is transfected. These errors make it impossible to include the requested experiment. However, we did analyze the effect of the siRNA treatments using multiplex dual reporter plasmids that just include one pathway luciferase reporter and the normalizer ELuc luciferase, compared to the same treatments performed with the multiplex hextuple luciferase reporter. In Supplemental Figure 16b-g we convincingly demonstrate that pathway activities measured by the multiplex hextuple luciferase vector (which measures all five activities at once) correlate with pathway activities measured by the five multiplex dual luciferase plasmids (which each measure a single activity). Hence, these experiments confirm that the observed measurements obtained by using the multiplex hextuple luciferase vector truly reflect the on-target as well as collateral effects of the different siRNA treatments and are not a consequence of cross-talk between the different response elements located on the same plasmid. It is important to note that if cross-talk could occur *in cis* between response elements located on the same plasmid, they might as well occur *in trans* between response element located on separate plasmids (cotransfected or solotransfected in the same cell). Fortunately, our results demonstrate that this doesn't seem to be the case.

- 2. Do the reporters reflect the activity of each pathway? The authors should measure the activity of the transcription factors reported in at least one other assay such as DNA-binding ability, nuclear translocation etc. for each of the transcription factor complexes and assess potential collateral effects in these assays also.**

Although we understand the reviewer's concern, we should point out that in our work we have included DNA response elements that have been previously described (see Supplementary Table 1 with appropriate references) and are used in a multitude of examples in the literature (hundreds if not thousands of papers). These DNA reporters have been characterized with DNA-binding assays, mutagenesis of the sites, etc. and are well accepted by the scientific community as pathway reporters for each of the selected pathways.

- 3. In Figure 5 all pathway outputs should be shown in all knockdowns. There is no data shown for SMAD2 knockdown for instance. C-fos knockdown did not work and in fact reduced the effect of c-jun knockdown on c-fos levels. This should be repeated with an efficacious siRNA.**

We have now included a panel in Figure 5 in which we analyze the effect of the SMAD2 siRNA knockdown. In order to do this, we pre-treated the A549 cells with recombinant TGF- β ligand, to activate a pathway that otherwise demonstrates, very low, basal activity in these cells (see Figure 5c). We have also selected three new siRNAs for c-Fos and cotransfected these in a mix to effectively knockdown the expression of c-Fos (see Figure 5f).

Minor points:

- 1. The final plasmid should be sequenced**

Essential DNA Parts, vectors and the final multiplex luciferase vector were deposited in Addgene. Addgene has a high standard quality control processing pipeline that currently uses next-generation sequencing (NGS), obtaining full plasmid sequences. The integrity of our building blocks and the final vector has been

independently validated by Addgene, and vectors have been cleared for distribution, when the paper is accepted (see attached Deposit 76269 documentation).

2. Figure 1 legend needs n numbers and errors added

N numbers and errors were included in the legend.

3. Some figures are quoted to 5 decimal places (e.g., sup Fig3 supp table 1,2) which will overestimate the accuracy of the measurements.

Figures have been quoted to two decimal places. This was addressed in Supplementary Figures S3 and S4, and in Supplementary Table 1, and 2. Inverse matrix values in Figure S4 were quoted to four decimal places for appropriate accuracy of the calculations. Regression coefficients of determinations were quoted to three decimal places.

4. Supp Fig 7 remove “see e” and “see f”

This was fixed in Supplementary Figure 7.

Reviewer 2

Minor criticisms:

1. In the qPCR assays, it would be useful to include direct measure of the luciferase transcripts to compare with the enzyme activity. Only downstream targets, which are indirect readouts, are included at present.

The goal for the multi-luciferase assay is to use the luminescence signals of each luciferase as readouts for the changes in transcription factor-regulated gene expression. Therefore, increases or decreases of luminescence signals should be reflected in changes the expression of genes that are regulated by the specific transcription factors, measured by qRT-PCR. We believe it is not necessary for us to directly measure the luciferase transcript levels because we are using luciferase enzymatic activity, not its transcript, as reporter of each pathway. The multiluciferase vector would only be useful if the luminescence of each luciferase, not the luciferase transcripts, match the transcription factors' activities.

2. The use of the language "the CMV promoter" is slang, and is more correctly referred to as the hCMV major immediate early promoter or at least, the hCMV-IE1 promoter.

The reviewer correctly points out that the promoter should be better named in our manuscript. We have now referred to as the hCMV-IE1 promoter throughout the text, as suggested. We have also changed the nomenclature in Figure 1, 2, and 3, and Supplemental Figures 1, 7, 8, 9, 10, 11, 12, 13, and 14. We requested to Addgene the promotor renaming of plasmid 118048 from **pCMV_P** to **phCMV-IE1**, and of plasmids 118062, 118063, 118064, 118065, 118066, and 118067 to reflect the suggested hCMV-E1 nomenclature (see attached Deposit 76269 documentation).

3. In the paragraph of speculation of the potential applications of this method in the Discussion, it would be more useful to include a balanced description of the limitations of the method. For example, the assay is best applied to easily transfected lines. Another limitation is the size of the plasmid used (13.4 kb), as larger plasmids become more difficult to transfect successfully. It is also possible that there could be cross-talk between reporters (for example, through enhancer action) when incorporated on the same molecule, although this does not appear to be the case under the conditions shown in the study.

We apologize for not explaining more clearly these limitations in our manuscript. We have addressed the reviewer's points in the Discussion section of the paper by including a full paragraph discussing concerns and limitations as well as some solutions to those concerns and limitations (including references).

Reviewer 3

Major comments:

1. To control for all of these possibilities, in the experiment shown Fig 6 in particular, I would consider it essential to repeat the experiment, with all 6 luciferases under the control of the same promoter in order to exclude the possibility that the effect of each drug treatment may instead be attributable

to some post-transcriptional effect on one or more of the luciferase activities. Ideally this would occur by the same solotransfection method, but it would be acceptable to simply repeat the drug assays using each of the 6 CMV driven luciferases (Supp table 4) separately, in the wild type cell line, presented in an additional supplementary figure. In order to increase the robustness and reproducibility of that the wonderful technology they have developed, the authors might consider including a sentence in the discussion as to why it is important to control for such off-target effects in luciferase-based screens.

The reviewer pointed out an important limitation of the luciferase technology, which are the potential off-target effects of chemical compounds on luciferase activities. We have performed an experiment in which we assayed the six CMV-IE1 driven luciferases in the A549 cell line, with all the drugs in Figure 6, as suggested. We normalized the light emission with the total protein in each well for easier comparison between treatments. Results indicate that none of the drugs we used in our experiments affect the light emission of the luciferases. As a positive control, we included Pifithrin- α , a well-known *in vitro* and *in vivo* inhibitor of the activity of FLuc, which repressed 3.5 fold the activity of FLuc. We have referenced this experiment in the results section of the paper, and also in the discussion, we highlighted the importance of the verification of potential off-target effects in luciferase-based screens. This experiment is now included as Supplemental Figure 17 in our manuscript and the rest of supplemental figure numbers have been adjusted accordingly. Also, information about the total protein quantification using the bicinchoninic acid assay (BCA) is included in the Methods section of the manuscript.

Minor comments

1. **1c seems to be a replot of some of the same data presented in 1b rather than a separate experiment. If so this should be mentioned explicitly in the legend.**

The reviewer is right about this. A short sentence has been included in the legend of Figure 1 to mention it.

2. **Page 6, line 2 & abstract - many researchers use luciferases for longitudinal, live cell assays. The manuscript describes a destructive endpoint assay, as described in other parts of the manuscript. So as not to confuse the naive reader, it might be worth reiterating at this point that the various luciferase activities measured post-lysis. In the interests of clear communication, it might be worth also mentioning that the endpoint nature of the assay in the abstract.**

This was addressed in the abstract and in the description of the technique, as well as in the discussion section of the paper. We also suggested a solution for longitudinal studies in the discussion.

3. **Page 11, line 11 - the observation that coelenterazine-responsive luciferases are associated with high experimental error requires some discussion. I do not doubt the observation, but find it hard to rationalize, as may others.**

In our manuscript, the intention was not to imply that coelenterazine-responsive luciferases are associated with high experimental errors. We have observed, and demonstrated in Supplemental Figure 11, that the deconvolution of the coelenterazine-luciferases was not accurate when one of them was used as experimental standard. We have included a description of this issue in that sentence, to emphasize our statement, and to avoid possible misunderstanding as the reviewer highlighted.

4. **Page 12, line 5 - the authors are of course welcome to call this 'solotransfection' if they wish, but the term is unlikely to catch on because it doesn't distinguish what the authors have done, from the fairly standard practice of transfecting cells with single plasmids that express more than one protein. I suggest that they come up with a better name if they wish it to enter common parlance.**

We have thought about this a lot. We believe that the scientific community should clearly separate cotransfection methods from transfecting a single plasmid. Currently it is generally unclear what approach is used, although most of the times people do cotransfections (while calling it transfection). For that reason, we wanted to emphasize a new term, solotransfection, to separate transfection procedures transforming a single plasmid encoding the entire synthetic circuit (solotransfection), from the procedures that transform all constituents of the synthetic circuit as separate entities (cotransfection often referred to as transfection).

5. **siRNA controls - I accept that the siRNA knockdown data presented in this manuscript is showing proof-of-principle, but it is very unusual these days not to include scrambled or non-targeting**

siRNA sequences in order to control for non-specific effects that transfection and introduction of exogenous small RNAs into the cytoplasm has on cells - this may trigger dysregulation of the endomembrane system or an innate intracellular immune response, for example, which could interfere with the assay result. So far as I can discern, no siRNA controls are reported in this manuscript. It is not sufficient to report that these sequences have been validated previously, it is important to show that transfection of each cell line of interest with some non-relevant siRNA did not affect the activity of the hextuple reporter or if effects are observed in the controls, that these be quantified and taken into account. If the authors do not wish to perform siRNA controls, they should state explicitly that no siRNA control transfections were performed in the legend of figure 5, the methods section and the relevant portion of the results.

The reviewer points an unintended omission in the methods section of our manuscript. The MISSION® siRNA Universal Negative Control #1 (Sigma-Aldrich) was used as non-targeting siRNA, and all targeting siRNA were compared to this negative control. This information is now included in the methods section.

6. CMV promoter - Are the authors confident that the CMV promoter is constant under all conditions. In our experience its activity can vary greatly between different cell culture conditions. In the context of a high content screen, for example, it was not quite clear to me how some exogenous perturbation that affects transcriptional regulation at the CMV promoter would be taken into account. Apologies if I have missed this.

We understand the reviewer's concerns. We should mention that we tested several promoters under varying experimental conditions and found that from all promoters tested (hCMV-IE1, hEF1A, PGK, SV40, data not shown in the paper), the CMV promoter was the most consistent in expression levels. For example, across a number of 96-well experiments using the A549 lung cancer cell line we observed a level of CMV driven expression ranging 1 to 2×10^5 RLU/s. Furthermore, for a particular set of experiments all using the same culture conditions, the internal CMV driven control luciferase remains consistent. It may vary given other experimental conditions, but since the comparisons would all be within similarly treated conditions this should not be an issue when analyzing the data as the variation should be similar in any set of experiments. We have included a mentioning about the choice of the hCMV-IE1 promoter in the paper.

Also, an error was detected in Figures 1i, Supplemental Figure 1a-b, and 7 where "Fold change" was indicated in the graphs. This has been changed to "Log2 Fold change" for consistency with the rest of the figures in the manuscript. We have also re-arranged some panels in the figures, for consistence related to gene order amongst them. In addition, we updated all human gene names to the current recommendation (e.g., the gene p53 is now TP53).

Finally, all raw data underlying any of the reported averages in graphs and charts is included in the "source data" file, named "Source Data". In this file, raw data is included for figures 1b-h, 2b-d, 2f-h, 4b-d, 5b-f, 6b-e, 6g, 6i-l, supplementary figures 5b-e, 6b-e, 8b-d, 9b-d, 11b-d, 12b-d, 15a-e, 16a, 16c-g, 17a-d, 18b-d, 19, 20a-d, and supplementary tables 1 and 2. A sentence indicating "Source data are provided as a Source Data file" is now mentioned at the end of the legend for each of these figures and tables.

REVIEWERS' COMMENTS:

Reviewer #1 (Remarks to the Author):

I thank the authors for addressing my comments and congratulate them on an excellent study. It is good to see that when transfected as single reporter genes the effects of knockdowns seen on the multiplex reporter are reproduced. Regarding the measurement of other activities of the pathway such as nuclear translocation, DNA binding etc I accept that the reporters used have been previously well validated for the pathways that they report on but my point was that when the authors observe collateral effects of perturbation of one pathway on another then doing these kind of assays would reveal at what point in the signaling pathway the collateral effects occur. I think it would be good to mention this point in the discussion in the paper.

Minor errors. Page 13 (see supplemental Figure 16a-e... should read "see supplemental Figure 15, Figure 16a-e...)

Figure 5f + missing on the JNK luciferase reading in the first panel

Gareth Inman

Reviewer #3 (Remarks to the Author):

This is a great contribution to the bioassays literature. The authors have adequately addressed all my concerns about their initial submission. I recommend this manuscript be accepted for publication, and I hope the technology catches on.

John O'Neill

Please find enclosed the response to the reviewers of the revision of our manuscript “**Examining multiple cellular pathways at once using multiplex hextuple luciferase assaying**” by Alejandro Sarrion-Perdigones, Lyra Chang, Yezabel Gonzalez, Tatiana Gallego-Flores, Damian Young, and Koen Venken.

Reviewer 1:

Minor errors:

1. **Page 13 (see supplemental Figure 16a-e... should read "see supplemental Figure 15, Figure 16a-e...)**
This was corrected.
2. **Figure 5f + missing on the JNK luciferase reading in the first panel**
This was corrected.